# *Akkermansia muciniphila* identified as key strain to alleviate gut barrier injury through Wnt signaling pathway

Xin Ma[1,2], Meng Li[1,2], Yuanyuan Zhang[1], Tingting Xu[1], Xinchen Zhou[1,2], Mengqi Qian[1], Zhiren Yang[1,2], Xinyan Han[1,2]*

[1]Key Laboratory of Animal Nutrition and Feed Science in East China, Ministry of Agriculture, College of Animal Sciences, Zhejiang University, Hangzhou, China; [2]Hainan Institute of Zhejiang University, Yongyou Industry Park, Yazhou Bay Sci-Tech City, Sanya, China

## eLife Assessment

The work by Han and collaborators describes **valuable** findings on the role of Akkermansia muciniphila during ETEC infection. If confirmed, these findings will add to a growing list of beneficial properties of this organism. The strength of the evidence used to justify the conclusions in the manuscript is **solid**, as the analyses broadly support the claims with only minor weaknesses.

*For correspondence:
xyhan@zju.edu.cn

Competing interest: The authors declare that no competing interests exist.

**Abstract** As the largest mucosal surface, the gut has built a physical, chemical, microbial, and immune barrier to protect the body against pathogen invasion. The disturbance of gut microbiota aggravates pathogenic bacteria invasion and gut barrier injury. Fecal microbiota transplantation (FMT) is a promising treatment for microbiome-related disorders, where beneficial strain engraftment is a significant factor influencing FMT outcomes. The aim of this research was to explore the effect of FMT on antibiotic-induced microbiome-disordered (AIMD) models infected with enterotoxigenic *Escherichia coli* (ETEC). We used piglet, mouse, and intestinal organoid models to explore the protective effects and mechanisms of FMT on ETEC infection. The results showed that FMT regulated gut microbiota and enhanced the protection of AIMD piglets against ETEC K88 challenge, as demonstrated by reduced intestinal pathogen colonization and alleviated gut barrier injury. *Akkermansia muciniphila* (*A. muciniphila*) and *Bacteroides fragilis* (*B. fragilis*) were identified as two strains that may play key roles in FMT. We further investigated the alleviatory effects of these two strains on ETEC infection in the AIMD mice model, which revealed that *A. muciniphila* and *B. fragilis* relieved ETEC-induced intestinal inflammation by maintaining the proportion of Treg/Th17 cells and epithelial damage by moderately activating the Wnt/β-catenin signaling pathway, while the effect of *A. muciniphila* was better than *B. fragilis*. We, therefore, identified whether *A. muciniphila* protected against ETEC infection using basal-out and apical-out intestinal organoid models. *A. muciniphila* did protect the intestinal stem cells and stimulate the proliferation and differentiation of intestinal epithelium, and the protective effects of *A. muciniphila* were reversed by Wnt inhibitor. FMT alleviated ETEC-induced gut barrier injury and intestinal inflammation in the AIMD model. *A. muciniphil*a was identified as a key strain in FMT to promote the proliferation and differentiation of intestinal stem cells by mediating the Wnt/β-catenin signaling pathway.

## Introduction

Growing evidence suggests that gut microbiota and its metabolites play important roles in modulating host health (*Canfora et al., 2019*; *Ghosh et al., 2021*; *Zheng et al., 2021*). The most drastic exposure that leads to intestinal microbiota dysbiosis is antibiotic treatment, which kills commensal micro-organisms and inhibits these microbiota-mediated immune defense (*Andremont et al., 2021*; *Buffie and Pamer, 2013*). The disturbance of intestinal microbiota may lead to disruptions of the gut barrier and increase the susceptibility of the host to pathogenic microbes (*Flint et al., 2012*; *Witkowski et al., 2020*). Several clinical studies have demonstrated that early-life exposure to antibiotics leads to gut microbiota dysbiosis that can impair host immune system maturation (*An et al., 2014*; *Nguyen et al., 2020*). Furthermore, the negative impact of antibiotics on gut microbiota in early life may even last for long periods throughout a lifetime, increasing the risk of gut diseases (*Francino, 2015*). Enterotoxigenic *Escherichia coli* (ETEC) K88 infection is a common cause of diarrhea in humans and animals (*Dubreuil, 2021*). When ETEC adheres to intestinal epithelial cells, virulence factors interact with specific receptors to induce intestinal barrier injury and intestinal inflammatory responses (*Yu et al., 2018*). Fecal microbiota transplantation (FMT) is a therapeutic intervention for reconstructing gut microbiota in gastrointestinal inflammatory diseases (*Borody et al., 2019*; *Colman and Rubin, 2014*). Since FMT has been successful in treating *Clostridioides difficile* infection and maintaining the gut barrier, the emerging investigation has been focused on other diseases (*Quraishi et al., 2017*). However, whether FMT could remedy against ETEC K88 infection in antibiotic-induced gut microbiota-disordered (AIMD) piglets is still unknown. Simultaneously, the underlying mechanism of FMT and the gut microbes that confer its efficacy remain unclear.

Intestinal stem cell renewal is vital for the maintenance of the gut barrier (*Beumer and Clevers, 2021*). Mammalian intestinal epithelial cells are renewed approximately every 3–5 d with the migration of undifferentiated intestinal cells from the crypt to the top of the villi, accompanied by the differentiation of different types of epithelial cells (*Luo et al., 2022*). The proliferation and differentiation of intestinal epithelium driven by intestinal stem cells, mainly crypt-base columnar cells expressing R-Spondin receptor Lgr5, is a necessary process for repairing intestinal barrier injury to prevent pathogen invasion (*Yan et al., 2017*). *Sato et al., 2009* successfully cultured isolated intestinal stem cells into three-dimensional intestinal organoids with villous and crypt-like structural morphology for the first time. Intestinal organoids contain various terminally differentiated cell types, including intestinal stem cell, tuft cell, absorptive cell, enteroendocrine cell, goblet cell, and Paneth cell, which have more advantages over single cell lines for in vitro study of intestinal regeneration (*Sprangers et al., 2021*). Intestinal organoid is an ideal model for studying the interaction between intestinal epithelium and microorganisms in vitro, which can eliminate the complexity of animal models (*Yin et al., 2019*). In traditional basal-out intestinal organoid models, the intestinal epithelium was in the interior of an organoid spheroid, which restricted the interaction between intestinal epithelium and gut microbiota. *Co et al., 2019* first established the intestinal organoid model with polarity reversal in humans and mouse, which optimizes the shortcomings of microinjection in terms of heterogeneous exposure concentrations and durations, as well as destruction of organoid walls, and evaluated the infection effects of the invasive pathogens with different polarity-specific patterns.

In the present study, we first investigated the effects of FMT on ETEC K88 infection in AIMD neonatal piglets, then by using bacterial 16 S rDNA sequencing analysis, we identified *Akkermansia muciniphila* (*A. muciniphila*) as a potential microbe to alleviate ETEC-induced intestinal barrier injury (*Figure 1A*). We further hypothesized that *A. muciniphila* could relieve intestinal barrier injury and intestinal inflammation by modulating the proliferation and differentiation of intestinal epithelium, and verified this hypothesis using in vivo mice experiments as well as in vitro porcine intestinal organoid models (*Figure 1B–C*). Collectively, our results may provide the theoretical basis that *A. muciniphila* is a promising method to repair intestinal barrier damage and a new strategy for the precise application of *A. muciniphila* in livestock production.

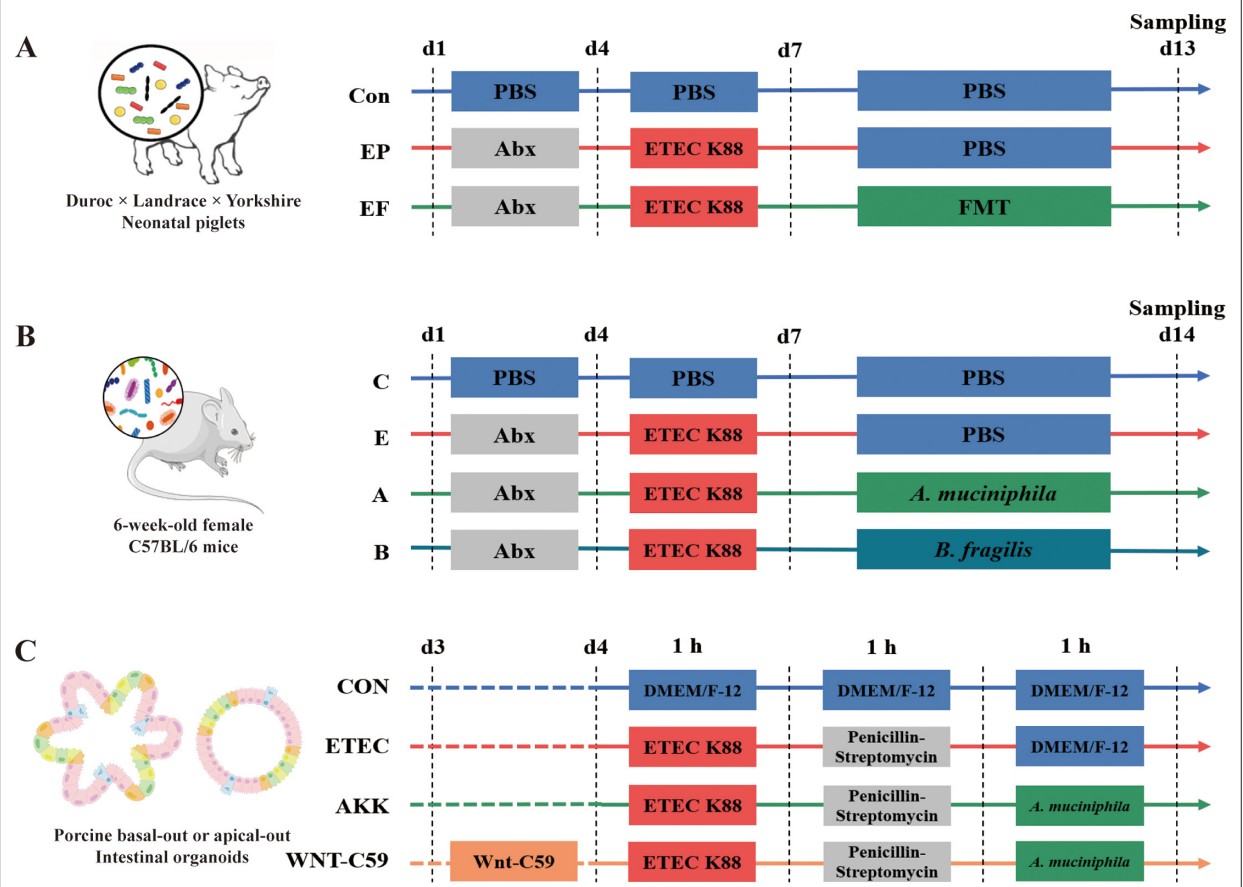

**Figure 1.** Experimental designs of the present study. (**A**) Schematic diagram of the fecal microbiota transplantation (FMT) administered to the antibiotic-induced microbiome-disordered (AIMD) piglets infected with Enterotoxigenic *Escherichia coli* (ETEC) K88. (**B**) Schematic diagram of the alleviatory effects of *A. muciniphila* and *B. fragilis* on ETEC K88 infection in mice. (**C**) Schematic diagram of the alleviatory effects of *A. muciniphila* on ETEC K88 infection in porcine basal-out and apical-out intestinal organoids.

## Results

### FMT enhanced the protection of AIMD piglets against ETEC K88 challenge

ETEC K88 infection is a leading cause of diarrhea in humans and animals. We set out to explore whether FMT is protected against the ETEC K88 challenge. The piglets were randomly assigned to three groups: the control piglets (Con group), the ETEC K88 infected piglets (EP group), and the ETEC K88 infected piglets with FMT treatment (EF group). As shown in *Figure 2A and B*, decreased ADG and increased diarrhea scores were observed in AIMD piglets on day 3 and day 6 after ETEC K88 infection, and these symptoms were relieved by FMT treatment. We next detected the pathogen colonization in mucosal tissues and found that FMT significantly decreased the copies of ETEC K88 in the jejunal and colonic mucosa (*Figure 2C*). To address whether FMT could protect the intestinal barrier against ETEC K88 infection, intestinal barrier integrity, and inflammatory cytokine levels were detected. FMT decreased the expression of proinflammatory cytokines (*TNF-α*, *IL-1β*, and *IL-6*) and increased levels of anti-inflammatory cytokines (*IL-10* and *TGF-β1*) in the jejunum (p<0.01, *Figure 2D*). Jejunal scanning electron microscopy showed that the villi (*Figure 3A* top) in ETEC K88 infected piglets were short and coarse, and there existed obvious rod-shaped bacterial adhesion on the surface of microvilli (*Figure 3A* bottom), while FMT reversed villi and microvilli injury caused by ETEC K88. In addition, the reduced mRNA and protein levels of the tight junctions (ZO-1, claudin, and occludin) and adherence junctions (β-catenin and E-cadherin) caused by ETEC K88 challenge were also reversed by FMT treatment in the jejunum (*Figure 3B–C*). We then evaluated the expression of Mucin 2 (MUC2) in the jejunum using immunofluorescence staining. As shown in *Figure 3D*, ETEC K88

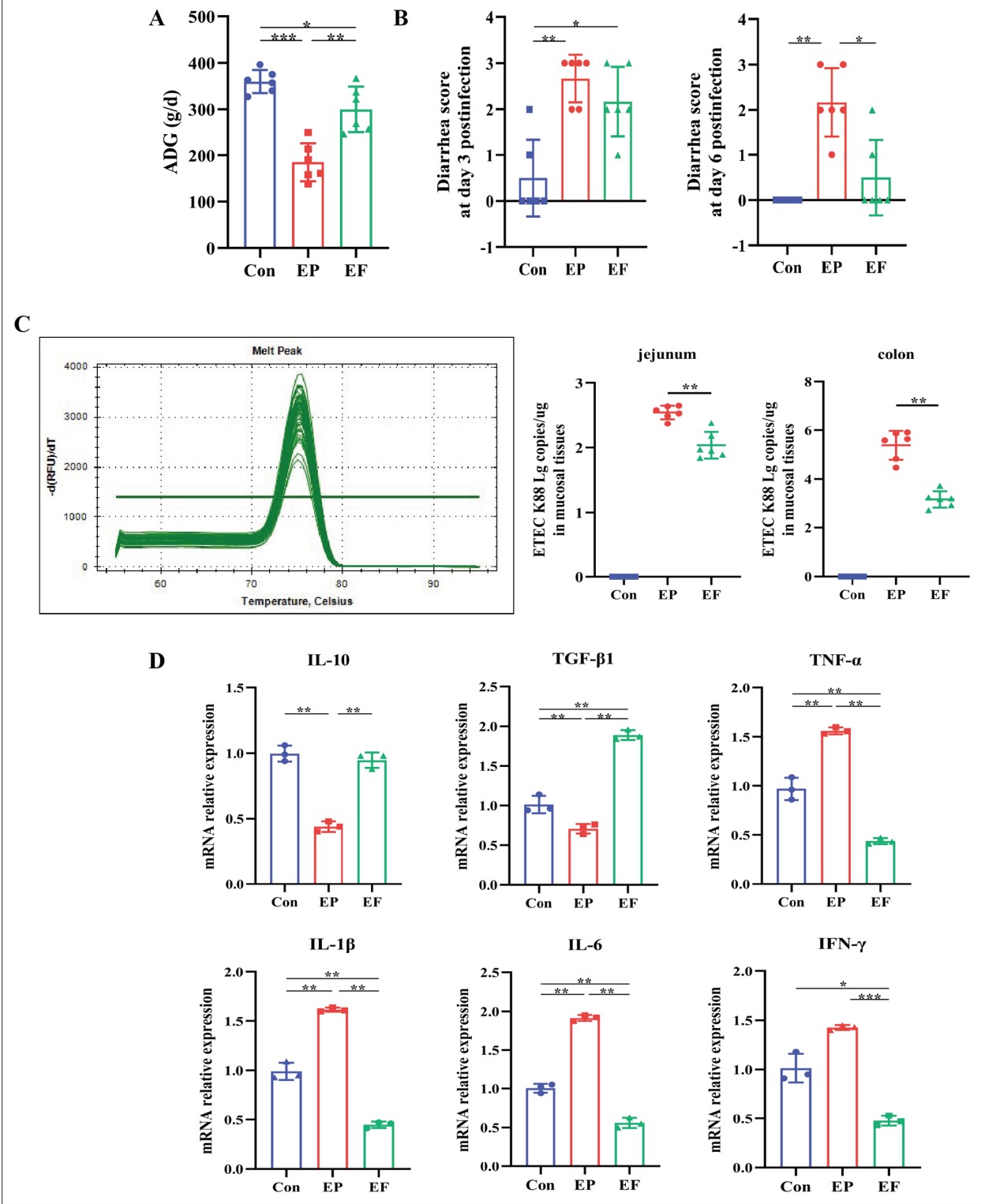

**Figure 2.** Fecal microbiota transplantation (FMT) improved the growth performance and inflammatory response of AIMD piglets infected with Enterotoxigenic *Escherichia coli* (ETEC) K88. (**A**) Average daily weight gain (ADG) in the Con, EP, and EF groups. (**B**) Diarrhea score at days 3 and 6 post-infection. (**C**) The colonization of ETEC K88 in the jejunum and colon. The melting curve demonstrated that only the ETEC K88 could be amplified by the primers we used. (**D**) The relative mRNA expression of cytokines in the jejunum. Con: control group, EP: ETEC K88 +PBS group, EF: ETEC K88 +FMT group. Data are expressed as the mean ± SD. *p<0.05, **p<0.01, ***p<0.001.

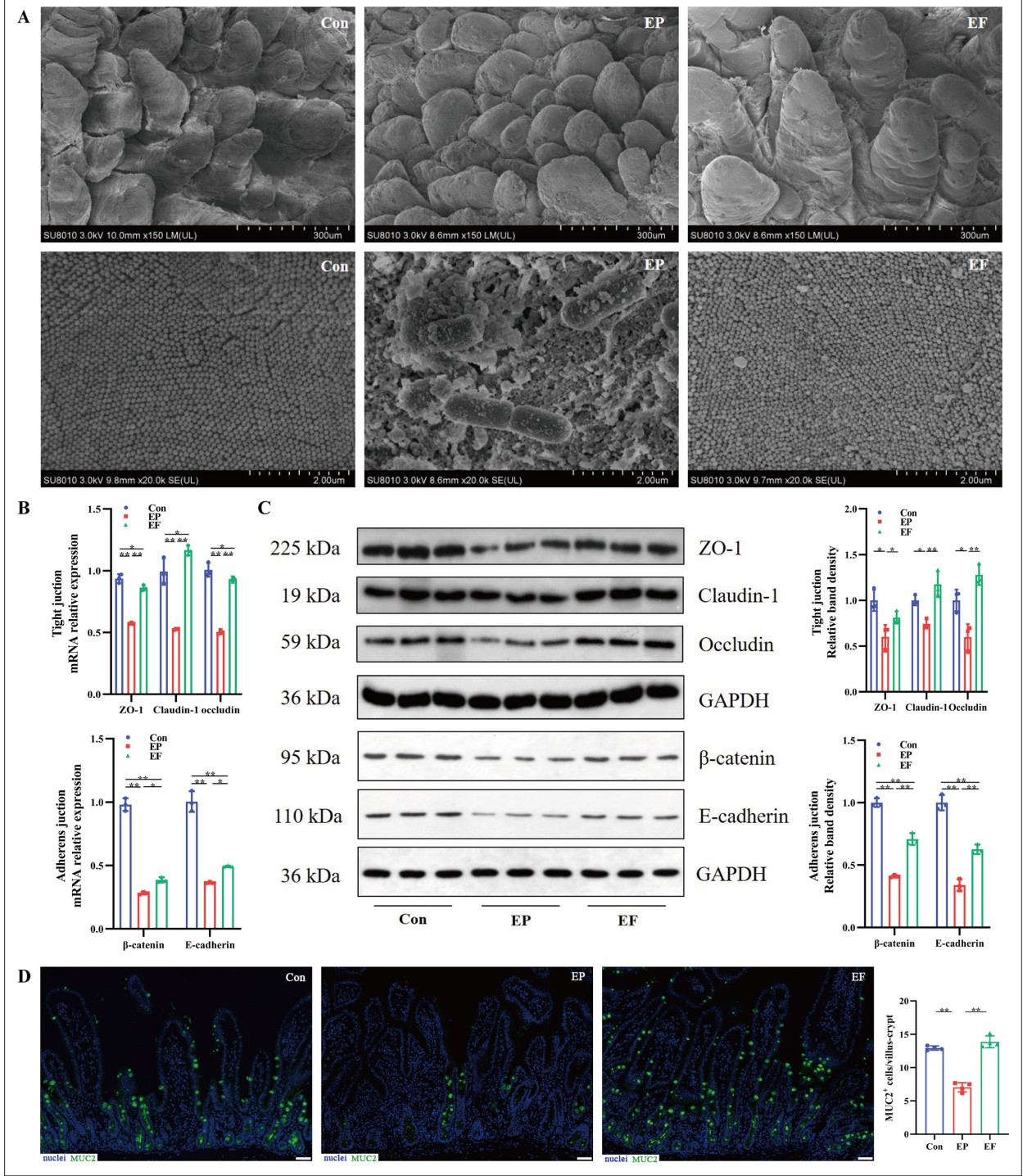

**Figure 3.** Fecal microbiota transplantation (FMT) improved the intestinal morphology and barrier function in antibiotic-induced microbiome-disordered (AIMD) piglets infected with Enterotoxigenic *Escherichia coli* (ETEC) K88. (**A**) Scanning electron microscopy (SEM) images of villi (top) and microvilli (bottom) in the jejunum (scale bars = 300 or 2 μm). (**B**) The relative mRNA expression of tight junction proteins (ZO-1, claudin, and occludin) and adheren junctions (β-catenin and E-cadherin) in the jejunum. (**C**) The relative protein expression of tight junction proteins (ZO-1, claudi, and occludin) and adheren junction proteins (β-catenin and E-cadherin) in the jejunum. (**D**) Immunofluorescence images of Mucin 2 (MUC2, green) in the jejunum (scale bars = 50 μm). Con: control group, EP: ETEC K88 +PBS group, EF: ETEC K88 +FMT group. Data are expressed as the mean ± SD. *p<0.05, **p<0.01. Data shown are representative of at least two independent experiments.

The online version of this article includes the following source data for figure 3:

**Source data 1.** The image files of scanning electron microscope.

*Figure 3 continued on next page*

*Figure 3 continued*

**Source data 2.** The image files of Western blot.

**Source data 3.** The image files of immunofluorescence (MUC2).

infection reduced the expression of MUC2, which was significantly increased following FMT administration. These results indicated that FMT enhanced the protection of AIMD piglets against the ETEC K88 challenge.

## *A. muciniphila* and *B. fragilis* were identified as two key strains in FMT

We next assessed the variation of gut microbiota and SCFA metabolites in ETEC K88-challenged piglets. The alpha diversity indexes Chao1 and Observed species were increased after FMT treatment (*Figure 4—figure supplement 1A–B*). The PCoA and NMDS showed that gut microbiota between the two groups clustered separately (*Figure 4—figure supplement 1C–D*). The Anosim analysis ($R=0.396$, p=0.02) and UPGMA clustering analysis also indicated that FMT altered the structure of the gut microbiota in ETEC K88-challenged piglets (*Figure 4A–B*). LEfSe analysis verified that *B. fragilis* and *Akkermansia* were changed significantly with an upper 4 LDA score after FMT administration (*Figure 4C*). We compared the OTU sequence of *Akkermansia* in the National Center for Biotechnology Information database by using the Blast method based on the Best Hit algorithm, and the result was *A. muciniphila*. Furthermore, FMT significantly increased the production of SCFA metabolites, including acetic acid, propionic acid, butyric acid, isobutyric acid, isovaleric acid, and hexanoic acid (*Figure 4D*). These results demonstrated that FMT regulated gut microbiota and SCFA metabolites in ETEC K88-challenged piglets. *A. muciniphila* and *B. fragilis* were identified as two strains that may play key roles in FMT.

## *A. muciniphila* and *B. fragilis* maintain intestinal barrier function of ETEC-induced mice

We further investigated whether these two strains exerted protective effects on ETEC infection in the AIMD mice model. Mice were randomly assigned to four groups: the control mice (C group), the ETEC K88 infected mice (E group), the ETEC K88 infected mice treated with *A. muciniphila* (A group) and the ETEC K88 infected mice treated with *B. fragilis* (B group). As shown in (*Figure 5A*, *Figure 5—figure supplement 1A*), ETEC K88 infection led to weight loss and intestinal morphology damage, which was relieved by *A. muciniphila* and *B. fragilis* treatment. Correspondingly, scanning electron microscopy showed that *A. muciniphila* and *B. fragilis* increased the height and number of microvilli and ameliorated the distribution of intercellular junction in the jejunum and colon (*Figure 5B*). In addition, the relative mRNA and protein expressions of MUC2 were significantly increased in the jejunum of *A. muciniphila* and *B. fragilis*-treated mice (*Figure 5C*). However, only *A. muciniphila* could upregulated MUC2 protein expression in the colon of treated mice (*Figure 5C*). Furthermore, we investigated the diffusion of ETEC K88 in feces and organs after infection. ETEC K88 infection surged the number of *Escherichia coli* in the feces, jejunum, colon, liver, spleen, and kidney (*Figure 5—figure supplement 1B*). *A. muciniphila* and *B. fragilis* treatment dramatically decreased the amount of ETEC K88 in these tissues (*Figure 5—figure supplement 1B*), indicating a reduced translocation of ETEC K88 in the tissues and organs. The concentration of FD4 in the serum has been commonly used as an indicator of intestinal permeability. As shown in *Figure 5—figure supplement 1C*, *A. muciniphila*, and *B. fragilis* significantly decreased serum FD4 concentration, indicating *A. muciniphila* and *B. fragilis* enhanced intestinal barrier integrity.

As demonstrated in *Figure 6A*, ETEC K88 decreased the relative mRNA expression of anti-inflammatory cytokines *IL-10* and *TGF-β* and increased levels of pro-inflammatory cytokines *TNF-α*, *IL-1β*, *IL-6* and *IFN-γ*, which was reversed in A and B groups, indicating that *A. muciniphila* and *B. fragilis* ameliorated ETEC K88-induced intestinal inflammation. Intestinal inflammatory response is closely related to immune cells. Treg cells secrete anti-inflammatory cytokines IL-10 and TGF-β to alleviate intestinal inflammation. Conversely, Th17 cells produce pro-inflammatory cytokines, such as IFN-γ and IL-17, which aggravate intestinal inflammation. The dynamic balance of Treg and Th17 cells plays an important role in intestinal immunity. Therefore, we used flow cytometry to analyze the proportion of the immune cells isolated from jejunal lamina propria in the innate and adaptive immune system. Treg

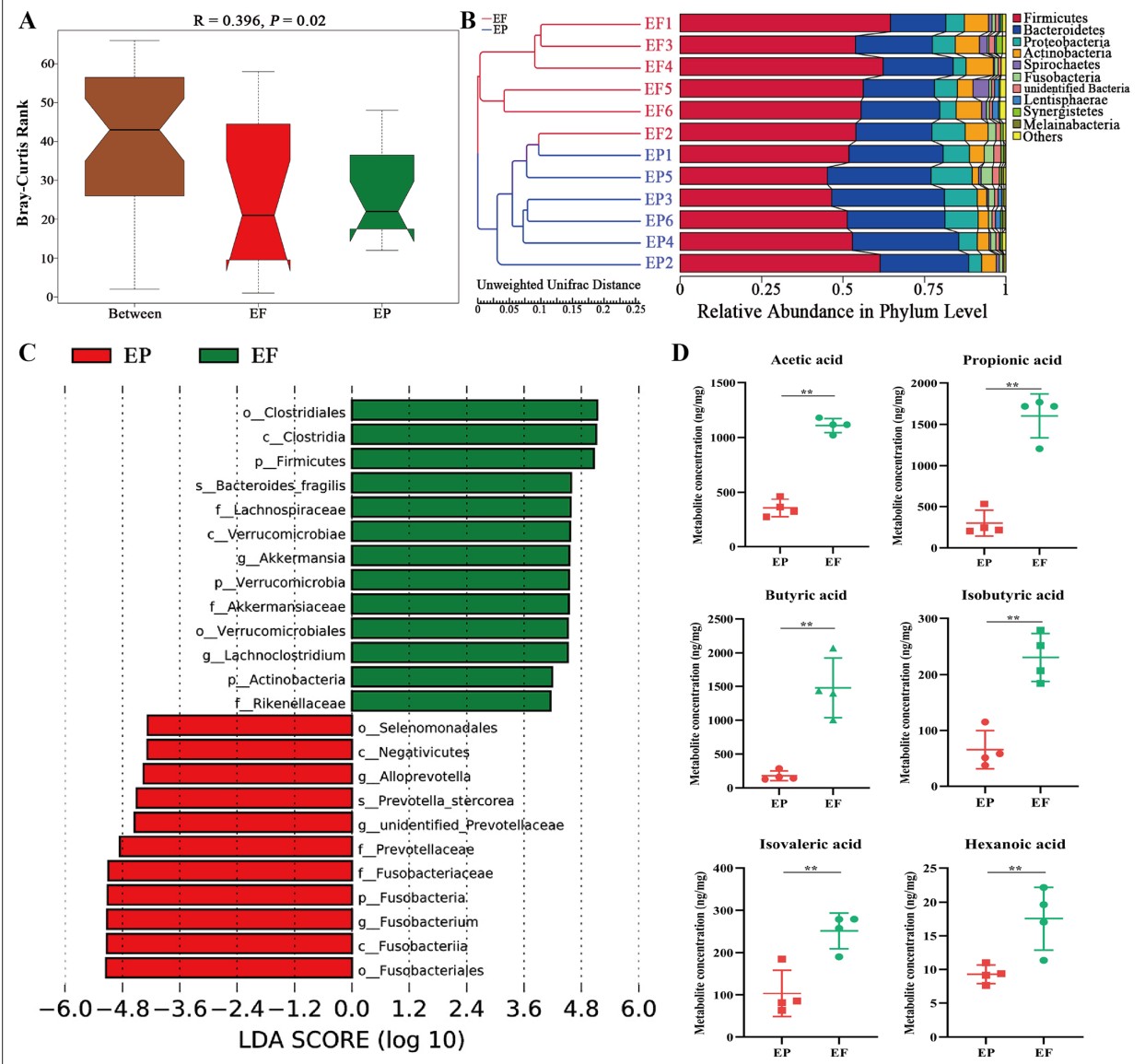

**Figure 4.** Fecal microbiota transplantation (FMT) changed the composition of gut microbiota and the concentration of short-chain fatty acid (SCFAs) in antibiotic-induced microbiome-disordered (AIMD) piglets infected with Enterotoxigenic *Escherichia coli* (ETEC) K88. (**A**) Anosim analysis. (**B**) UPGMA clustering analysis based on Unweighted unifrac distance. (**C**) Linear effect size (LEfSe) analysis with linear discriminant analysis (LDA) score >4. (**D**) Quantification of short-chain fatty acid (SCFA) metabolites (acetic acid, propionic acid, butyric acid, isobutyric acid, isovaleric acid, and hexanoic acid). EP: ETEC K88 +PBS group, EF: ETEC K88 +FMT group. Data are expressed as the mean ± SD. *p<0.05, **p<0.01. Data shown are representative of at least two independent experiments.

The online version of this article includes the following figure supplement(s) for figure 4:

**Figure supplement 1.** Fecal microbiota transplantation (FMT) altered the structure of gut microbiota in antibiotic-induced microbiome-disordered (AIMD) piglets infected with Enterotoxigenic *Escherichia coli* (ETEC) K88.

cells were labeled with CD25+Foxp3+ cells, Th17 cells were labeled with CD4+RORγt+ cells, and mature dendritic cells were labeled with CD86+CD11C+ cells. ETEC K88 infection significantly decreased the proportion of Treg cells and mature dendritic cells and increased the proportion of Th17 cells, while oral administration of *A. muciniphila* or *B. fragilis* significantly reduced the proportion of Th17 cells and increased the proportion of mature dendritic cells (*Figure 6B*). Moreover, compared with E group, the proportion of Treg cells in A group was significantly increased (p<0.01), and the proportion of Treg cells in the B group had an increasing trend (p=0.59), indicating FMT maintained the proportion of Treg/Th17 cells in ETEC K88-infected mice.

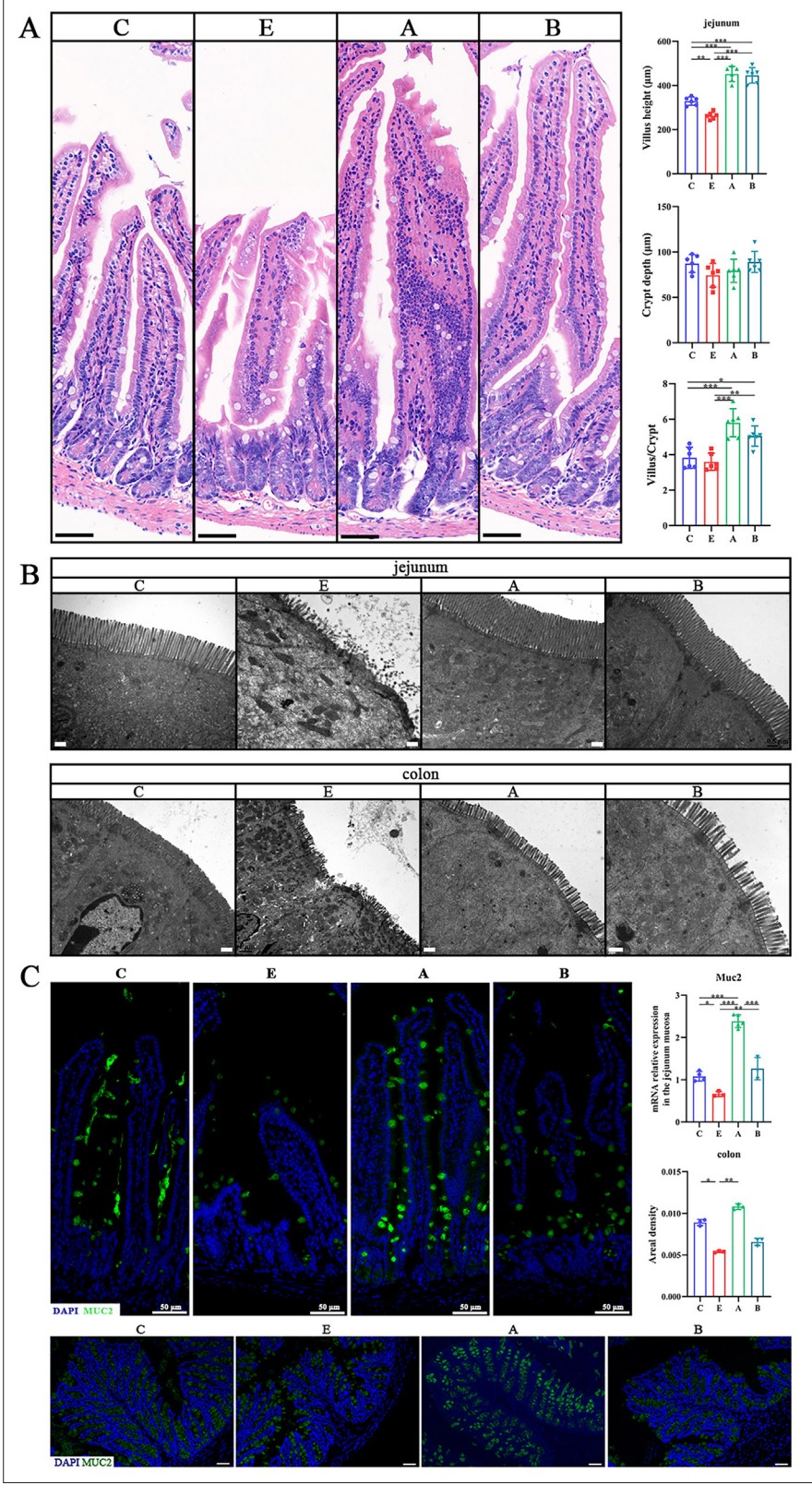

**Figure 5.** *A. muciniphila* and *B. fragilis* improved the intestinal morphology of mice infected with Enterotoxigenic *Escherichia coli* (ETEC) K88. (**A**) Hematoxylin-eosin (H&E)-stained images of jejunum (scale bars = 50 μm) and the histograms of villus height, crypt depth, and the ratio of villus height to crypt depth. (**B**) Transmission electron microscopy (TEM) images of jejunum and colon (scale bars = 0.5 μm). (**C**) Immunofluorescence images of Mucin

*Figure 5 continued on next page*

*Figure 5 continued*

2 (MUC2, green) in the jejunum and colon (scale bars = 50 µm). (C: control group, E: ETEC K88 +PBS group, A: ETEC K88 +*A*). *muciniphila* group, B: ETEC K88 +*B. fragilis* group. Data are expressed as the mean ± SD. *p<0.05, **p<0.01, ***p<0.001. Data shown are representative of at least two independent experiments.

The online version of this article includes the following source data and figure supplement(s) for figure 5:

**Source data 1.** The image files of hematoxylin-eosin (H&E) staining.

**Source data 2.** The image files of transmission electron microscope.

**Source data 3.** The image files of immunofluorescence (MUC2).

**Figure supplement 1.** *A. muciniphila* and *B. fragilis* improved the weight and intestinal permeability of mice infected with Enterotoxigenic *Escherichia coli* (ETEC) K88.

## *A. muciniphila* accelerated intestinal epithelial proliferation

Since oral administration of *A. muciniphila* and *B. fragilis* mitigated intestinal mucosal injury, we set out to explore their effects on the proliferation and differentiation of intestinal epithelium. ETEC K88 infection prominently decreased *Ki67* and *Lyz* mRNA expression and the number of Ki67+ cells and Lyz+ Paneth cells in the crypt of the jejunum (**Figure 7A and B**). *A. muciniphila* and *B. fragilis* increased *Ki67* and *Lyz* mRNA expression and the number of Ki67+ cells. However, only *A. muciniphila* enhanced the number of Lyz+ Paneth cells, suggesting that *A. muciniphila* accelerated the proliferation and differentiation of intestinal epithelial cells. Lgr5-labeled active intestinal stem cells are the driving force of intestinal epithelial proliferation and differentiation after intestinal injury. ETEC K88 infection significantly increased the relative mRNA expression of *Lgr5*, *Wnt3*, *Axin2*, *Ctnnb1* in jejunum mucosa and had no significant impact on *Notch1* and *Hes1* expression (**Figure 7C**). *A. muciniphila* and *B. fragilis* inhibited the upregulation of *Lgr5*, *Wnt3*, *Axin2,* and *Ctnnb1* at transcriptional level caused by ETEC K88 infection and had no significant effect on *Notch1* and *Hes1* levels. Western blot analysis also showed that *A. muciniphila* and *B. fragilis* inhibited the upregulation Lgr5, Wnt3, Active β-catenin, c-Myc, and CyclinD1 at protein level, and the effect of *A. muciniphila* was better than that of *B. fragilis* (**Figure 7D**). These results implied that *A. muciniphila* regulated the expression of intestinal stem cells in ETEC K88-challenged mice and regulated Wnt/β-catenin signaling pathway rather than Notch signaling pathway.

We next isolated jejunal crypt from mice and explore whether *A. muciniphila* or *B. fragilis* influenced the ability of intestinal stem cells growing into intestinal organoids ex vivo. In ETEC K88-infected mice, the crypts formed into undifferentiated spherical organoids, while in *A. muciniphila* or *B. fragilis*-treated mice, the crypts generated mature organoids with bud structure (**Figure 8A**). We statistically analyzed the surface area and forming efficiency of the intestinal organoids. As shown in **Figure 8B**, the organoids derived from ETEC K88-infected mice had a smaller surface area and lower forming efficiency than those from control mice. *A. muciniphila* treatment led to a larger surface area and higher forming efficiency of the intestinal organoids. The surface area of intestinal organoids in B group was larger than that in E group, while there was no significant difference in forming efficiency between these two groups. Meanwhile, the relative mRNA expression of genes related to the proliferation and differentiation at the intestinal organoid level was consistent with the results of jejunal mucosa in vivo (**Figure 8C**). These results suggested that *A. muciniphila* accelerated intestinal epithelial proliferation and regulated Wnt/β-catenin signaling pathway ex vivo.

## *A. muciniphila* protected intestinal organoids against ETEC infection via Wnt signaling

To explore the effects and mechanisms of *A. muciniphila* on ETEC K88 infection in vitro, the porcine intestinal organoids were infected with ETEC K88 (ETEC group), and then co-cultured with *A. muciniphila* (AKK group). Wnt inhibitor Wnt-C59 was added to explore whether *A. muciniphila* acted through Wnt/β-catenin signaling pathway (WNT-C59 group), and porcine intestinal organoids without any treatment were used as a blank control (CON group). The relative mRNA expression of *villin*, *ZO-1*, *Ki67*, *Lyz*, *MUC2*, *Lgr5*, *Wnt3a*, *β-catenin* and the fluorescence intensity of villin, Ki67, Lgr5, Wnt3a, and β-catenin in basal-out intestinal organoids were significantly down-regulated upon ETEC exposure (**Figure 9A–G**). *A. muciniphila* upregulated Lgr5, Wnt3a, and β-catenin expression in ETEC-infected organoids, while these effects were inhibited by Wnt-C59 treatment (**Figure 9A–G**). The

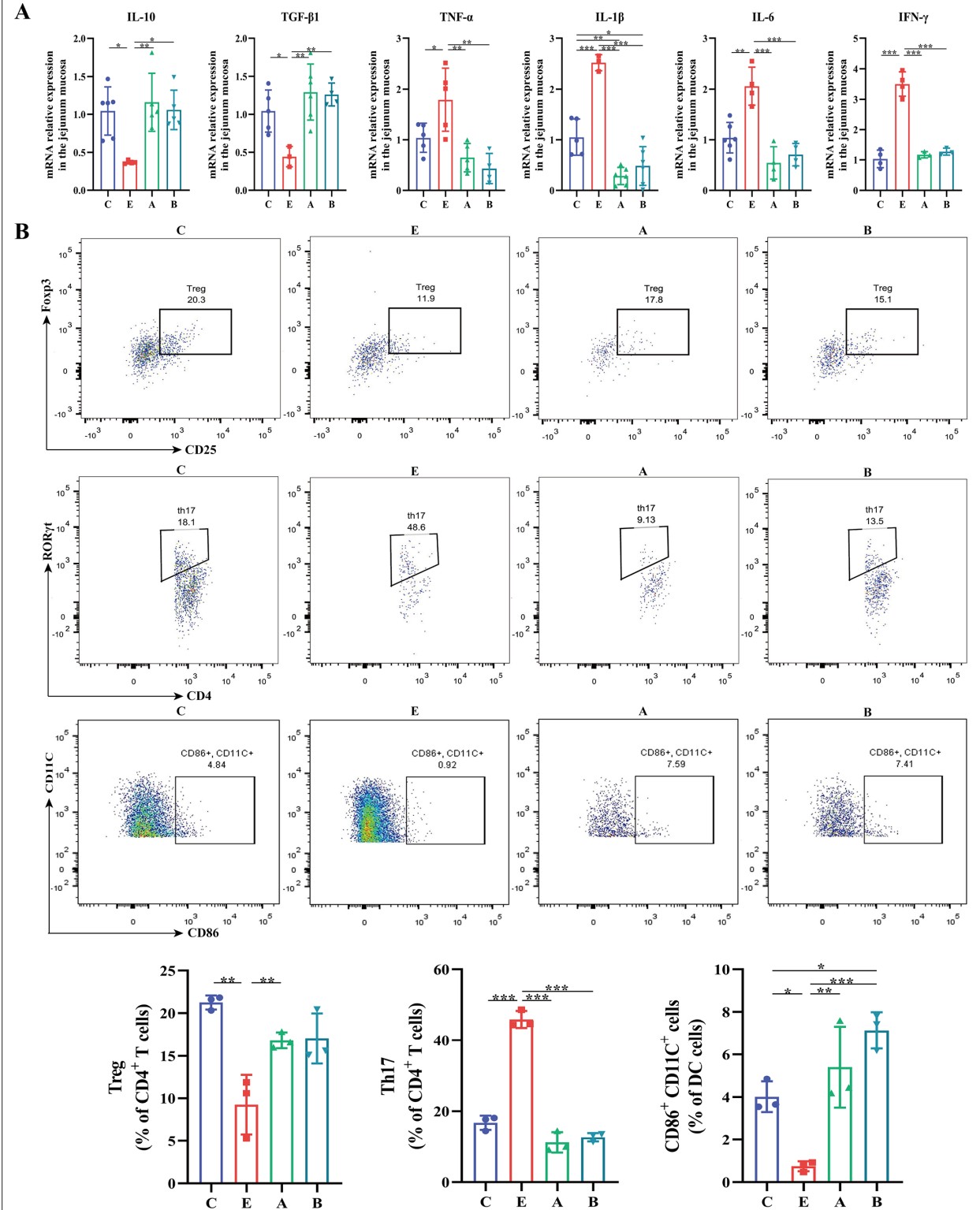

**Figure 6.** *A. muciniphila* and *B. fragilis* regulated the expression of inflammatory cytokines and the balance of Treg and Th17 cells in mice infected with Enterotoxigenic *Escherichia coli* (ETEC) K88. (**A**) The relative mRNA expression of cytokines in the jejunum. (**B**) Flow cytometric dot plots and proportions of Treg (CD25+Foxp3+), Th17 (CD4+RORγt+) and mature dendritic cells (CD86+CD11C+) in jejunal lamina propria. (C: control group, E: ETEC K88 +PBS group, A: ETEC K88 +*A). muciniphila* group, B: ETEC K88 +*B. fragilis* group. Data are expressed as the mean ± SD. *p<0.05, **p<0.01, ***p<0.001. Data shown are representative of at least two independent experiments.

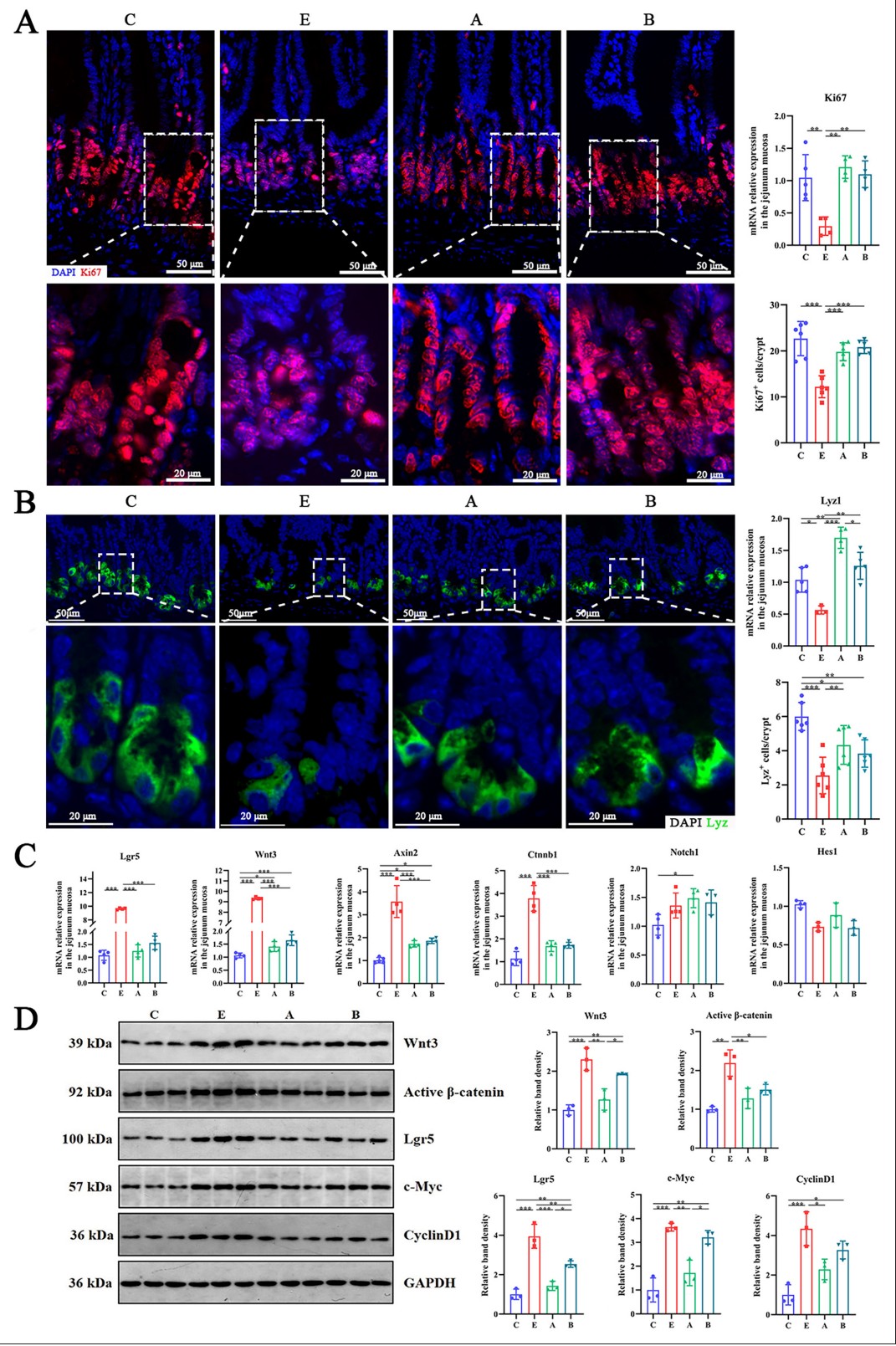

**Figure 7.** *A. muciniphila* and *B. fragilis* enhanced the number of Ki67 cells and Lyz Paneth cells in jejunal crypt of mice infected with Enterotoxigenic *Escherichia coli* (ETEC) K88 by moderately activating the Wnt/β-catenin pathway. (**A**) The mRNA expression and immunofluorescence staining images of intestinal epithelial cell proliferation marker Ki67 (red). (**B**) The mRNA expression and immunofluorescence staining images of Paneth

*Figure 7 continued on next page*

*Figure 7 continued*

marker Lyz (green). (**C**) The relative mRNA expression of *Lgr5, Wnt3, Notch1* and other genes in jejunum mucosa. (**D**) The relative protein expression of Wnt/β-catenin pathway and its target genes in jejunum mucosa. (C: control group, E: ETEC K88 +PBS group, A: ETEC K88 +*A). muciniphila* group, B: ETEC K88 +*B. fragilis* group. Data are expressed as the mean ± SD. *p<0.05, **p<0.01, ***p<0.001. Data shown are representative of at least two independent experiments.

The online version of this article includes the following source data for figure 7:

**Source data 1.** The image files of immunofluorescence (KI67 and LYZ).

**Source data 2.** The image files of Western blot.

results suggested that *A. muciniphila* increased the number of intestinal stem cells and activated Wnt signaling, but did not promote the proliferation and differentiation of basal-out gut organoids. In basal-out intestinal organoid models, the intestinal epithelium was in the interior of organoid spheroid, which restricted the interaction between intestinal epithelium and *A. muciniphila*. Therefore, we reversed the polarity of basal-out intestinal organoids by removing the Matrigel from the culture system and suspending intestinal organoids in the culture (*Figure 10—figure supplement 1A*). In apical-out intestinal organoids, the apical surface was outward to make it accessible to *A. muciniphila*. Under confocal microscopy, the F-actin of basal-out intestinal organoid was located in the inside of the spheroids, while the F-actin of the apical-out intestinal organoid was outside the spheroids, indicating that the polar reversal intestinal organoid model was successfully constructed (*Figure 10—figure supplement 1B*). We conducted FD4 permeation test in apical-out intestinal organoid model, and co-cultivation with *A. muciniphila* restored the epithelial barrier integrity of organoids challenged by ETEC K88 (*Figure 10A and H*). *A. muciniphila* not only increased the mRNA expression of *ZO-1, Lyz,* and *MUC2* in the ETEC K88 infected apical-out intestinal organoids, but also up-regulated the protein expression of villin, Wnt3a, β-catenin, and the number of Lgr5[+] and Ki67[+] cells (*Figure 10B-G, I*). Moreover, *A. muciniphila* also relieved the inflammatory response of the ETEC K88 infected apical-out intestinal organoids, as determined by decreasing proinflammatory cytokines, inducing *TNF-α, IL-1β, IL-6,* and *IFN-γ*, and increasing anti-inflammatory cytokines *IL-10* and *TGF-β* (*Figure 10J*). However, the moderating effect of *A. muciniphila* on the apical-out intestinal organoids disappeared after Wnt-C59 exposure. Taken together, *A. muciniphila* accelerated the proliferation and differentiation of the intestinal stem cells and ameliorated the intestinal barrier injury and inflammation induced by ETEC K88. *A. muciniphila* protected intestinal organoids against ETEC infection via Wnt signaling.

## Discussion

The homeostasis of gut microbiota is a major contributor to host health. Intestinal dysbiosis increases intestinal colonization of antibiotic-resistant and pathogenic bacteria. Antibiotic treatment is the most drastic exposure leading to gut dysbiosis and pathogen invasion (*Andremont et al., 2021*). Early-life exposure to antibiotic leads to gut microbiota dysbiosis and impair host immune system maturation, which may even last for long periods throughout life (*Nguyen et al., 2020*). Researchers have made a lot of attempts to prevent or inhibit antibiotic-induced dysbiosis. Our previous study showed that FMT alleviates early-life antibiotic-induced gut microbiota dysbiosis and mucosa injuries in a neonatal piglet model (*Ma et al., 2021*). Our present study aimed to investigate whether FMT increases early-life antibiotic-induced neonatal piglet against pathogen invasion. ETEC K88 is one of the most common pathogens in humans and animals and causes intestinal inflammation and diarrhea symptoms (*Sun and Kim, 2017*). Studies demonstrated that ETEC K88 infection led to pro-inflammatory cytokine upregulation and intestinal mucosal barrier injury (*Wang et al., 2020*; *Xie et al., 2021*). In ETEC-induced neonatal piglets, ZO-1 and Claudin-2 levels in ileum of were significantly decreased after ETEC K88 infection (*Xie et al., 2021*). In accordance with the above results, the present study showed that ETEC K88 infection increased the expression of pro-inflammatory cytokines and reduced the expression of intercellular junction proteins in AIMD piglets. Since FMT has been successful in treating *Clostridioides difficile* infection and maintaining gut barrier, emerging investigation has been focused on other diseases (*Cammarota et al., 2017*). *Geng et al., 2018* demonstrated that FMT diminished the inflammatory response and the destruction of epithelial integrity in piglets challenged with lipopolysaccharide. In the present study, we found that FMT dampened ETEC K88-induced pro-inflammatory

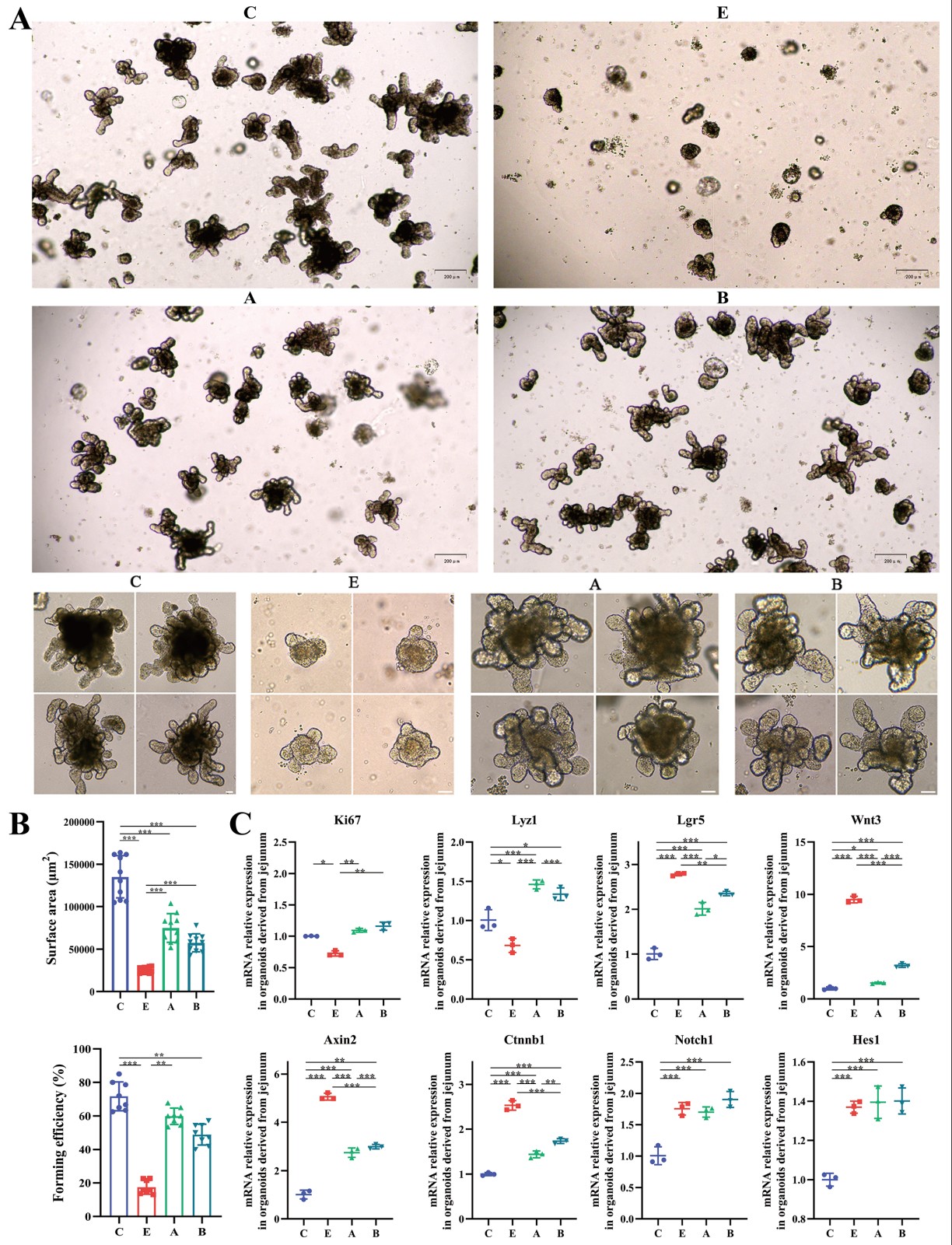

**Figure 8.** *A. muciniphila* and *B. fragilis* promoted the activity of Lgr5$^+$ intestinal stem cells as proved by the growth of intestinal organoids.
(**A**) Morphology of ex vivo culture of crypts isolated from the jejunum of mice in the C, E, A, and B groups (Day 5; 4x, scale bars = 200 µm; 20x, scale bars = 50 µm). (**B**) The surface area and forming efficiency of intestinal organoids. (**C**) The relative mRNA expression of genes related to the proliferation and differentiation of the intestinal organoids. (C: control group, E: ETEC K88 +PBS group, A: ETEC K88 +*A). muciniphila* group, B: ETEC K88 +*B. fragilis* group. Data are expressed as the mean ± SD. *p<0.05, **p<0.01, ***p<0.001. Data shown are representative of at least two independent experiments.

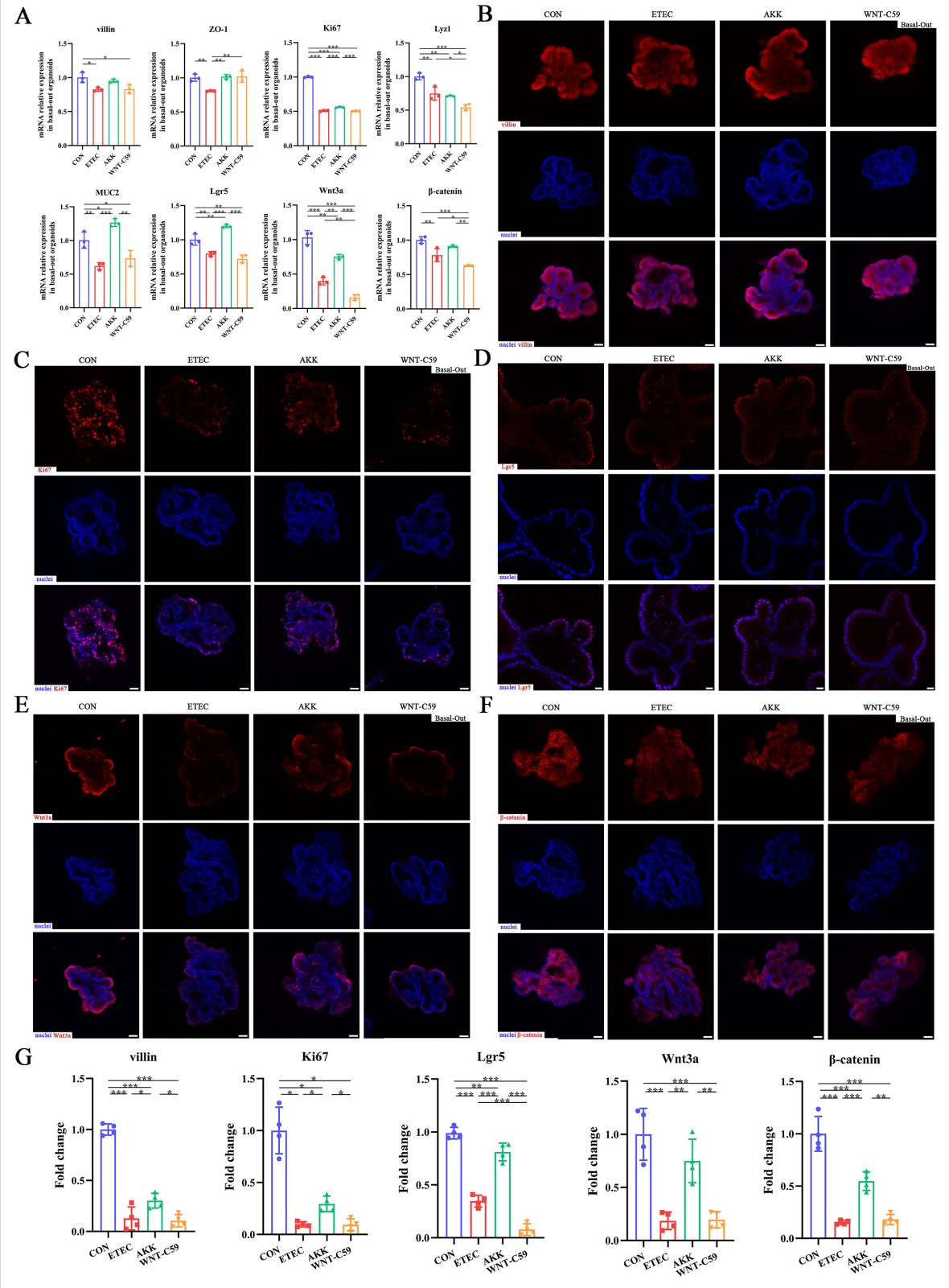

**Figure 9.** *A. muciniphila* protected the Lgr5+ intestinal stem cell and activated the Wnt/β-catenin signaling pathway of the Enterotoxigenic *Escherichia coli* (ETEC) K88-induced basal-out intestinal organoids. (**A**) The relative mRNA expression of *villin*, *ZO-1*, *Ki67*, *Lyz1*, *MUC2*, *Lgr5*, *Wnt3a* and *β-catenin* genes. (**B**)-(**F**) Immunofluorescence images of villin (scale bars = 50 µm), Ki67 (scale bars = 50 µm), Lgr5 (scale bars = 20 µm), Wnt3a (scale bars = 50 µm), and β-catenin (scale bars = 50 µm). (**G**) Fold change of the mean fluorescence intensity. CON: control group, ETEC: ETEC K88 +DMEM/F-12 group, AKK:

*Figure 9 continued on next page*

*Figure 9 continued*

ETEC K88 +*A. muciniphila* group, WNT-C59: Wnt-C59 +ETEC K88+*A. muciniphila* group. Data are expressed as the mean ± SD. *p<0.05, **p<0.01, ***p<0.001. Data shown are representative of at least two independent experiments.

The online version of this article includes the following source data for figure 9:

**Source data 1.** The image files of immunofluorescence.

cytokines upregulation and improved the intestinal morphology. In addition, the number of ETEC K88 in jejunal and colonic mucosa was significantly decreased following FMT, which was consistent with the observation of jejunal microvilli under SEM, showing the absence of rod-shaped bacterial adhesion in FMT treatment. ETEC infection was reported to reduce the MUC2 expression by destroying goblet cells or directly degrading MUC2, thereby damaging the integrity of the intestinal epithelial barrier (*Luo et al., 2014*). Our results demonstrated that FMT enhanced the expression of MUC2 and increased the expression of ZO-1, claudin, occludin, β-catenin, and E-cadherin at mRNA and protein levels, indicating that FMT improved the intercellular junctions between intestinal epithelial cells and restored the integrity of the intestinal mucosal barrier to reduce the colonization and invasion of ETEC K88.

Intestinal microbiota plays a key role in maintaining gut mucosal barrier and resistance to pathogen invasion (*Chelakkot et al., 2018*). Antibiotic-induced gut microbiota-disorders have been reported to disrupt gut barrier and increase the susceptibility of host to pathogenic microbes (*Wlodarska et al., 2011*). So, we further probed the changes of gut microbiota in AIMD piglets infected with ETEC K88. The results showed that FMT regulated gut microbiota and SCFAs metabolism in ETEC K88-challenged piglets. LEfSe analysis manifested that *B. fragilis* and *A. muciniphila* were the key differential bacteria between three groups. *A. muciniphila* utilizes mucin as the only carbon and nitrogen source for its growth in the gut, and its metabolite SCFAs can facilitate MUC2 secretion by goblet cells to maintain the dynamic balance of intestinal mucus layer (*Ottman et al., 2017*). *A. muciniphila* negatively correlated with many diseases, including diabetes, obesity, and inflammatory bowel diseases (*Belzer and de Vos, 2012*; *Castro-Mejía et al., 2016*; *Plovier et al., 2017*). It has been reported that *A. muciniphila* effectively inhibited intestinal inflammatory response and elevated epithelial barrier function (*Everard et al., 2013*; *Li et al., 2016*; *Zhang et al., 2018*). Meanwhile, *Bacteroides fragilis* is one of the symbiotic anaerobes within the mammalian gut and is also an opportunistic pathogen which often isolated from clinical specimens (*Sun et al., 2019*). Although it was initially thought to be pathogenic, in the long-term evolution process, *Bacteroides fragilis* colonized in the gut has established a friendly relationship with the host, which is an essential component for maintaining the health of the host, especially for obesity, diabetes, and immune deficiency diseases (*Troy and Kasper, 2010*). *B. fragilis* could decompose mucin by lipid-anchored enzymes and interact with intestinal epithelial cells, thereby altering intestinal permeability and repairing intestinal epithelial barrier (*Donaldson et al., 2016*; *Hsiao et al., 2013*). *Deng et al., 2016* reported that *B. fragilis* altered macrophage phenotype and increased the phagocytosis of macrophages to restrain pathogens colonization. Hence, we speculated that FMT increased the relative abundance of *A. muciniphila* and *B. fragilis* in the intestine and alleviate intestinal barrier injury and intestinal inflammation caused by ETEC K88 infection.

Since *A. muciniphila* and *B. fragilis* were identified as two strains that may play key roles in FMT, we further investigate whether *A. muciniphila* and *B. fragilis* exerted protective effect on ETEC K88 infection in AIMD mice model. We found that *A. muciniphila* and *B. fragilis* maintained normal gut morphology after ETEC K88 infection, repaired the integrity of intestinal mucosal barrier as evidenced by the reduced FD4 concentration in serum, colonized in the intestine and inhibit the translocation of ETEC K88, and reduced the intestinal inflammation, which may have been attributed to the increased proportion of Treg cells and mature dendritic cells and the decreased proportion of Th17 cells. These results are consistent with the protective role of *A. muciniphila* and *B. fragilis* in dextran sulphate sodium salt induced enteritis model (*Bian et al., 2019*; *Wang et al., 2021*; *Zhai et al., 2019*). Intestinal stem cell renewal is vital to maintaining the gut barrier. Ki67 is widely expressed in proliferating cells and presents a reliable indicator of cell proliferation activity (*Zheng et al., 2010*). Ki67 reaches its peak during the division stage with short half-life period and easy degradation (*Zheng et al., 2010*). In our study, *A. muciniphila* and *B. fragilis* increased the number of Ki67[+] cells and the mRNA expression of *Lyz*, suggesting that *A. muciniphila* and *B. fragilis* accelerated the proliferation and differentiation of intestinal epithelium. Paneth cells not only produce antimicrobial peptides and express key niche

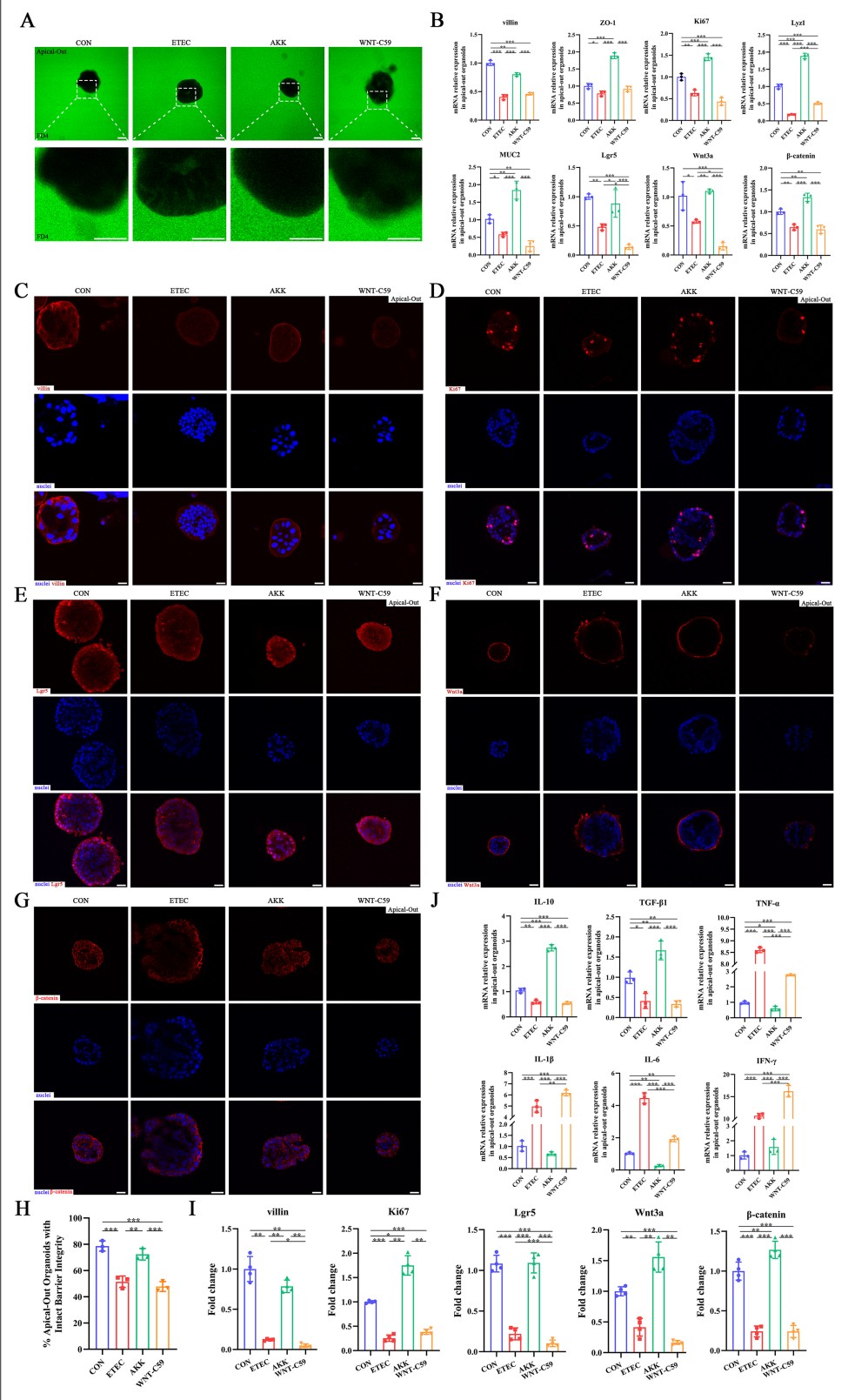

**Figure 10.** *A. muciniphila* alleviated the intestinal epithelial injury of the Enterotoxigenic *Escherichia coli* (ETEC) K88-induced apical-out intestinal organoids. (**A**) Confocal microscope visualization of the apical-out intestinal organoids incubated in FD4 solution (scale bars = 50 μm). (**B**) The relative mRNA expression of *villin*, *ZO-1*, *Ki67*, *Lyz1*, *MUC2*, *Lgr5*, *Wnt3a* and *β-catenin* genes. (**C**)-(**G**) Immunofluorescence images of villin, Ki67, Lgr5, Wnt3a,

*Figure 10 continued*

and β-catenin (scale bars = 20 μm). (**H**) Quantification of the apical-out intestinal organoids incubated in FD4 solution that have intact barrier integrity. (**I**) Fold change of the mean fluorescence intensity. (**J**) The relative mRNA expression of cytokines in the apical-out intestinal organoids. CON: control group, ETEC: ETEC K88 +DMEM/F-12 group, AKK: ETEC K88 +*A. muciniphila* group, WNT-C59: Wnt-C59 +ETEC K88+*A. muciniphila* group. Data are expressed as the mean ± SD. *p<0.05, **p<0.01, ***p<0.001. Data shown are representative of at least two independent experiments.

The online version of this article includes the following source data and figure supplement(s) for figure 10:

**Source data 1.** The image files of FD4.

**Source data 2.** The image files of immunofluorescence.

**Figure supplement 1.** The construction of polar reversal intestinal organoids.

signals that regulate stem cells, but also secrete activators that dynamically modulate Wnt/β-catenin signaling pathway (*Hou et al., 2020*). The Wnt/β-catenin signaling pathway is essential for the proliferation and differentiation of intestinal stem cells (*Ring et al., 2014*). When the Wnt receptors are activated, the multiprotein degradation complex is rapidly inhibited, which allows β-catenin to accumulate and subsequently migrate to the nucleus and binds to the transcription factors of T cell factor family and lymphoid enhancer-binding factor family to drive the expression of genes that regulate the stem cells, including Lgr5, c-Myc, and Cyclin D1 (*Niehrs, 2012*). The c-Myc and Cyclin D1 are cyclins that positively regulate cell cycle from the G1 phase to S phase, thus inducing cell cycle progression and promote cell proliferation (*Shi et al., 2015*). *Zhu et al., 2020* reported that *A. muciniphila* restored intestinal mucosal damage with the accelerated proliferation of intestinal epithelium through Wnt/β-catenin signaling pathway, thus relieving infection caused by *Salmonella pullorum* in chicks. In the present study, *A. muciniphila* moderately activated Wnt/β-catenin signaling pathway and maintain the suitable expression of Wnt, Active-β-catenin, Lgr5, c-Myc, and Cyclin D1. Consistent with the in vivo results, ex vivo results of mouse intestinal organoids also demonstrated that *A. muciniphila* repaired the damaged intestinal epithelium caused by the ETEC K88 infection by modulating the Wnt/β-catenin signaling pathway. Simultaneously, *B. fragilis* also avoid the overactivation of the Wnt/β-catenin signaling pathway caused by the ETEC K88 infection, while the impact of *B. fragilis* was not as obvious as *A. muciniphila*.

Intestinal organoid is an ideal model for studying the interaction between intestinal epithelium and microorganisms in vitro, which can eliminate the complexity of animal models and avoid the unicity of the traditional intestinal epithelial cells (*Yin et al., 2019*). In order to further verify whether *A. muciniphila* attenuated ETEC K88 infection through Wnt/β-catenin signaling pathway, basal-out and apical-out porcine intestinal organoid models was used. Notably, we found that the ETEC K88 infection down-regulated Wnt/β-catenin signaling pathway in vitro, which was inconsistent with the results in vivo. A recent study showed that the addition of 400 ng ml$^{-1}$ STp, an enterotoxin of ETEC, could suppress Wnt/β-catenin signaling pathway of porcine intestinal organoids (*Zhou et al., 2021*). It is speculated that the reason for the difference in results might be related to the difference in the processing time of the in vivo and in vitro treatment. In spite of the slight stimulation of *A. muciniphila* on the proliferation and differentiation of infected intestinal organoids, *A. muciniphila* up-regulated the Wnt/β-catenin signaling pathway and had protective effects on intestinal stem cells. In basal-out intestinal organoid models, the intestinal epithelium was in the interior of organoid spheroid, which restricted the interaction between intestinal epithelium and *A. muciniphila*. *Co et al., 2019* first established the intestinal organoid model with polarity reversal in human and mouse and evaluated the infection effects of the invasive pathogens with different polarity-specific patterns, *Salmonella enterica serovar Typhimurium* and *Listeria monocytogenes*, on polarity reversal intestinal organoids. Subsequently, (*Li et al., 2020*) first applied the porcine intestinal organoids with polarity reversal to investigate the infectivity and antiviral immune responses of porcine transmissible gastroenteritis virus. In addition, studies on host-pathogen interactions using polarity reversal intestinal organoids have also been reported in the sheep and chickens (*Nash et al., 2021*; *Smith et al., 2021*). But up to now, there are few reports on the interaction between host and symbiotic microorganism using polarity reversal intestinal organoids. Here, we utilized porcine polarity reversal intestinal organoids to make *A. muciniphila* directly interacting with the intestinal epithelium. Our results showed that *A. muciniphila* alleviated the injury of polarity reversal intestinal organoids induced by ETEC K88. In

agreement with the in vivo results, *A. muciniphila* restrain the intestinal inflammation caused by the ETEC K88 infection in polarity reversal intestinal organoids. Additionally, the alleviatory effect of *A. muciniphila* on ETEC K88 infection via the Wnt/β-catenin signaling pathway was confirmed with the addition of Wnt inhibitor Wnt-C59. The promoted proliferation and differentiation in the basal-out and apical-out intestinal organoids after co-culturing with *A. muciniphila* were reversed with Wnt-C59.

In conclusion, we found that FMT relieved intestinal inflammation and improved intestinal barrier function in AIMD piglets infected with ETEC K88. *A. muciniphila* was identified as a key strain in FMT to promote the proliferation and differentiation of intestinal stem cells by mediating the Wnt/β catenin signaling pathway. The study revealed the possible mechanism that FMT alleviates the intestinal barrier injury.

## Materials and methods

### Ethics statement

All animal experiments were performed according to the guidelines of the Animal Care Committee of Zhejiang University School of Medicine (permit number SYXK 2012–0178). The animal care protocol was approved by the Local Committee of Animal Use following the Guide for the Care and Use of Laboratory Animals (NIH publication 86–23 revised 1985).

### Preparation of fecal microbiota suspension and bacterial strains

The fecal microbiota suspension was prepared according to our previous method (*Ma et al., 2021*). In short, healthy adult Jinhua pigs that administrated medicinal feed additives or drugs for more than 3 mo were selected as fecal donors. The potential risks and infectious pathogens in donor pigs were determined by serological and stool testing to ensure the safety of the porcine FMT. Fresh fecal samples that collected from Jinhua pigs were transported in AnaeroPack (Mitsubishi Gas Chemical, Japan) on ice to laboratory within 2 hr. The 50 g fecal samples were homogenized in 250 mL sterile phosphate-buffered saline solution (PBS), filtered through sterile stainless-steel strainer, and then centrifuged at 4000 rpm for 10 min. The precipitate was resuspended in sterile PBS solution containing 10% sterile glycerol, and the fecal microbiota suspension was stored at –80°C. All the above preparation was performed in an anaerobic incubator (AW400SG anaerobic workstations; Electrotek, UK). All the facilities and tools for the preparation of the fecal microbiota suspension were sterilized prior to use. When processing the FMT, the fecal microbiota suspension was thawed in 37°C water bath. ETEC K88 serotype O149:K91:K88ac was purchased from the China Institute of Veterinary Drug Control (Beijing China), cultured in Luria-Bertani medium (Qingdao Hope Bio-Technology Company, China), and incubated at 37°C in a shaker with 250 rpm overnight. *A. muciniphila* ATCC BAA-835 and *Bacteroides fragilis* (*B. fragilis*) ATCC 25285 were purchased from Guangdong Microbial Culture Collection Center (GDMCC, Guangzhou, China). *A. muciniphila* was cultured in brain heart infusion media (OXOID, Thermo Fisher Biochemicals Ltd., UK) supplemented with 0.1% mucin (Sigma-Aldrich, USA). *B. fragilis* was cultured in trypticase soy broth (Qingdao Hope Bio-Technology Company, China) supplemented with 5% defibrinated sheep blood (Qingdao Hope Bio-Technology Company, China). These two anaerobic bacteria strains were incubated at 37°C in an anaerobic incubator with 10% $H_2$, 10% $CO_2$, and 80% $N_2$ (AW400SG anaerobic workstations; Electrotek, UK). The cultures were centrifuged for 10 min at a speed of 4000 rpm, and the pellets were then resuspended in either aerobic or anaerobic PBS.

### Animals

A total of 9 litters (9-11 piglets per litter) of Duroc×Landrace×Yorkshire ternary hybrid neonatal piglets (aged 5 d) with identical birth dates and parities were obtained from Anji Lvjiayuan Animal Husbandry Co. Ltd. (Zhejiang, China). Piglets were individually housed in pens with an appropriate environment of 24-26°C and 55–65% humidity. All newborn piglets were fed with artificial milk substitutes, in which the nutrients reached the requirements recommended by the National Research Council (NRC, 2012), and had ad libitum access to water.

A total of 130 female C57BL/6 mice aged 5 wk were obtained from Shanghai SLAC Laboratory Animal, Co., Ltd. (SCXK (Zhejiang) 2017-0005; Shanghai, China). Mice were housed in vinyl isolators

in a room with 23±1°C room temperature, 55-65% humidity, and 12 hr/12 hr light/dark schedule. All mice had free access to water and food.

## Experimental design

The piglets were randomly assigned to three groups (three litters per group): the control group (Con group), the ETEC K88 + PBS group (EP group), and the ETEC K88 + FMT group (EF group). The piglets in EP and EF groups were orally inoculated with 2 ml ampicillin (120 mg kg$^{-1}$) twice daily on day 1-3 of the experiment to disorder the intestinal resident microbiota, then orally inoculated with ETEC K88 (1 × 10$^9$ CFU ml$^{-1}$) suspended in 2 ml sterile PBS once daily on day 4–6 of the experiment. After ETEC K88 infection, the piglets in EF group received 1.5 ml fecal microbiota suspension once daily by oral gavage for 6 d, while the piglets in EP group received the same volume of sterile PBS. Meanwhile, the piglets in Con group were orally inoculated with the same volume of sterile PBS throughout the experiment. The doses of ampicillin and fecal microbiota suspension were based on a previous study by *Ma et al., 2021* . The weight of each piglet was measured at both the beginning and the end of the experiment to calculate the average daily gain (ADG), and diarrhea score (0, normal feces; 1, moist feces; 2, mild diarrhea; 3, severe diarrhea) was recorded throughout the experiment. Six piglets (2 piglets per litter) were picked from each group to be slaughtered at the end of the experiment (Day 13).

After acclimating for a week, mice were randomly assigned to four groups (n = 10 each): the control group (C group), the ETEC K88 + PBS group (E group), the ETEC K88 + *A. muciniphila* group (A group) and the ETEC K88 + *B. fragilis* group (B group). On days 1–3 of the experiment, the mice in E, A, and B groups were orally inoculated with 0.2 ml 15 mg ml$^{-1}$ ampicillin (Sigma-Aldrich, USA) twice daily to disorder the gut microbiota. On days 4–6 of the experiment, the mice in E, A, and B groups were infected with ETEC K88 (1×10$^9$CFU) suspended in 0.2 ml 0.1 M NaHCO$_3$ buffer (pH 9.0) by oral gavage once daily. Mice were fasted for 12 hr before infection. Meanwhile, the mice in C group were given the equal volume of sterile PBS on days 1–6 of the experiment. On days 7–13 of the experiment, live *A. muciniphila* (1×10$^6$CFU) and live *B. fragilis* (1×10$^8$CFU) suspended in 0.2 ml anaerobic sterile PBS were administered intragastrically once daily to the mice in A group and B group, respectively, while mice in C and E groups received the same volume of anaerobic sterile PBS. Mice were orally infused with 0.2 ml 40 mg ml$^{-1}$ 4 kDa fluorescein isothiocyanate-dextran (Sigma-Aldrich, USA) 4 hr before sampling. At the end of the experiment (Day 14), six mice in each group were randomly selected for ether anesthesia and neck dislocation.

## Culture and treatment of porcine intestinal organoid

The cryovial of porcine intestinal organoid was kindly provided by Professor Li Xiaoliang (Institute of Preventive Veterinary Medicine, College of Animal Sciences, Zhejiang University, Hangzhou, China). The frozen organoids were thawed in a 37°C-water bath for 2 min, and then added 1 ml of DMEM/F-12 (Gibco, USA) with 1% bovine serum albumin (BSA, BioFroxx, Germany) solution. The contents of the cryovial were mixed and transferred to the 15 ml conical tube containing 2 ml of DMEM/F-12 with 1% BSA solution, and then centrifuged at 200×g for 5 min at 4°C. The organoids were resuspended in equal volume of complete IntestiCult Organoid Growth Medium (06010, StemCell Technologies, Canada) and Matrigel Matrix (356231, Corning, USA). The 50 µl of the 500-crypt suspension was plated in the center of each well of the pre-warmed 24-well plate (3524, Corning, USA). The plate was placed at 37°C with 5% CO$_2$ for 10 min and then added 750 µl complete IntestiCult Organoid Growth Medium containing 100 µg ml$^{-1}$ penicillin/streptomycin (Invitrogen, USA) to each well. The culture medium was fully exchanged three times per week. For the treatment, the frozen organoids were passaged two times after thawing.

For generation of apical-out organoids, the procedure of reversing the polarity of organoids was performed according to a previously published protocol by *Co et al., 2021* . Briefly, after growing in Matrigel Matrix with growth medium for 7 d, the Matrigel-embedded organoids were gently dissolved with 500 µl cold 5 mM ethylene diamine tetraacetic acid (EDTA) in D-PBS and transferred to the 15 ml conical tube containing 10 ml of cold 5 mM EDTA. The tubes were incubated on a rotating platform at 4°C for 1 hr, and then centrifuged at 200×g for 3 min at 4°C. The organoids were resuspended in complete IntestiCult Organoid Growth Medium containing 100 µg ml$^{-1}$ penicillin/streptomycin and

transferred to ultra-low attachment 24-well plate (3473, Corning, USA). The suspended organoids were cultured at 37°C with 5% $CO_2$ for 3 d.

To induce intestinal damage, the culture medium of basal-out and apical-out organoids was changed to DMEM/F-12 containing ETEC K88 ($10^6$ CFU). After the invasion for 1 hr (*Qian et al., 2023*; *Xiao et al., 2022*), the organoids were transferred to DMEM/F-12 containing 200 µg ml$^{-1}$ penicillin/streptomycin for 1 h to kill residual extracellular bacteria. Then, the organoids were incubated in DMEM/F-12 containing *A. muciniphila* ($10^5$ CFU) for 1 hr to detect the repair effect of *A. muciniphila*. In addition, the organoids in WNT-C59 group were pretreated with the Wnt inhibitor Wnt-C59 (100 nM, AdooQ, China) for 24 hr prior to damage. After co-cultivation, the organoids were collected and resuspended in 4% paraformaldehyde (PFA, Servicebio, China) at 4°C or RNAiso Plus (Takara, Japan) at –80°C for further analysis. For epithelial barrier integrity, the apical-out organoids were resuspended in a solution of 4 kDa fluorescein isothiocyanate-dextran (FD4) (2 mg ml$^{-1}$) diluted in DMEM/F-12 without phenol red.

## Real-time quantitative PCR (RT-qPCR) analysis

Total RNA from intestinal tissue and organoids was extracted using TRIzol method according to the precise instructions and cDNA was synthesized using PrimeScript RT Master Mix (Takara, Japan). With cDNA as template, RT-qPCR was performed using TB Green *Premix Ex Taq* (Takara, Japan) on a Light-Cycler 480 System (Roche, Germany). The real-time primers used in this study are listed in *Supplementary file 1a*. The relative mRNA expression of target genes was statistically analyzed based on the $2^{-\triangle\triangle Ct}$ method.

## Bacterial load and intestinal permeability analyses

The bacterial load of ETEC K88 in piglets' jejunal and colonic mucosa was determined by the absolute quantification. In brief, the gene of ETEC K88 (Genbank Accession: M25302.1) was amplified by PCR (mK88-F: GGAATGGAAAGTTGGTACAGGTCTT, mK88-R: CCGGCATAAGATGCATTCACTTTC), and the PCR product was analyzed by agarose gel electrophoresis. The specific band was excised and recovered, connected with pGM-T vector, and transformed into high-efficiency chemoreceptor DH-5α cells. After culturing, the plasmid DNA was extracted and sequenced. The RT-qPCR analysis was performed as the description of section 2.5. The copy number of samples was calculated by the copy number of standard and the standard curve.

For the detection of ETEC K88 load in mice, fresh feces, jejunum and colon tissue, liver, spleen, and kidney were homogenized in sterile normal saline using a tissue homogenizer (Jingxin Industrial Development Co., Ltd., China), then the homogenates were diluted 10-fold continuously, and the dilutions were plated on MacConkey Agar (Qingdao Hope Bio-Technology Company, China). After incubating in an aerobic incubator at 37°C for 24 hr, the plates were counted three times. The results were presented as Lg CFU g$^{-1}$ of samples.

For intestinal permeability assessment in mice, the serum fluorescence was detected using a SpectraMax M5 plate reader (Molecular Devices, USA) at a 485 nm excitation wavelength and a 535 nm emission wavelength. The standard curve was established from the fluorescence of FD4 at concentrations of 0.005, 0.01, 0.02, 0.04, 0.08, 0.16 0.32, 0.64, 1.28, 2.56, 5.12, and 10.24 µg ml$^{-1}$ to calculate the FD4 concentration.

For detection of epithelial barrier integrity in organoids, the apical-out organoids in the FD4 solution were transferred to the slide and instantly imaged live by the laser scanning confocal microscope IX81-FV1000 (Olympus, Japan) with FV10-ASW software (Olympus, Japan).

## Histology and morphology analyses

Jejunum and colon tissues were collected, fixed in 4% PFA, dehydrated with gradient alcohol, and embedded in paraffin. The tissue sections were stained with hematoxylin-eosin (H&E) solution. The indexes related to the morphology of jejunum and colon were measured by using Image Plus (v6.0). Scanning electron microscopy (SEM) and transmission electron microscopy (TEM) were performed according to the procedure of the Bio-ultrastructure Analysis Lab of the Analysis Center of Agrobiology and Environmental Sciences, Zhejiang University. In brief, intestinal tissues were fixed in 2.5% glutaraldehyde, washed with PBS three times, postfixed with 1% OsO4 for 1 h, then dehydrated with gradient alcohol (30%, 50%, 70%, 80%, 90%, 95%, and 100%) for about 15 min at each step. For SEM,

the samples were dehydrated in Hitachi Model HCP-2 critical point dryer, coated with gold-palladium in Hitachi Model E-1010 ion sputter, and observed in Hitachi Model SU-8010 SEM. For TEM, the infiltrated samples were embedded in Spurr resin, sectioned in LEICA EM UC7 ultratome, and sections were stained by uranyl acetate and alkaline lead citrate, then observed in Hitachi Model H-7650 TEM.

## Immunofluorescence analysis

Jejunum tissues were fixed in 4% PFA, dehydrated, paraffin-embedded, and sectioned. After antigen retrieval and blocking, the sections were incubated in primary antibodies at 4°C overnight and secondary antibodies at room temperature for 50 min in the dark, followed by staining with DAPI at room temperature for 10 min in the dark. The fixed organoids were incubated in primary antibodies in blocking/permeabilization buffer at room temperature for 4 hr, and then incubated in secondary antibodies together with DAPI and phalloidin (Servicebio, China) in blocking/permeabilization buffer at room temperature for 2 hr in the dark. The fluorescence images were captured using a laser scanning confocal microscope IX81-FV1000 (Olympus, Japan) combined with FV10-ASW software (Olympus, Japan). The information of primary and secondary antibodies is shown in *Supplementary file 1b*.

## Western blot analysis

Total protein was extracted by radioimmunoprecipitation assay lysis buffer and protein concentration was determined by a BCA Assay Kit (Thermo Fisher Scientific, USA). Equal protein amounts (60 μg) were electrophoresed on 10% sodium dodecyl sulphate-polyacrylamide gel and the separated proteins were transferred onto the polyvinylidene difluoride membranes (Millipore, USA) The membranes were blocked with Tris-buffered saline with Tween (TBST) containing 5% BSA for 1 hr at room temperature, and incubated with the primary antibodies (*Supplementary file 1c*) in TBST containing 3% BSA overnight at 4°C. After several times washes in TBST, the membranes were incubated with secondary antibodies (*Supplementary file 1c*) in TBST containing 3% BSA for 1 hr at room temperature. The protein bands were visualized using SuperSignal West Dura Extended Duration Substrate (Thermo Fisher Scientific, USA), and quantified by Image J software.

## Microbiome sequencing analysis

Total DNA was extracted by CTAB method, and the purity and quality of DNA were detected by agarose gel electrophoresis. Specific primers 341 F (5'-CCTACGGGNGGCWGCAG-3') and 805 R (5'-GACTACHVGGGTATCTAATCC-3') were used for bacterial PCR amplification of 16S rDNA V3-V4 hypervariable region. The PCR products were purified using AMPure XP (Beckman Coulter Genomics, USA), and verified with the Illumina sequencing primer set by using Phusion HF Taq Polymerase (New England Biolabs, USA). The PCR products were sequenced on Illumina HiSeq 2500 platform. The low-quality part of reads was cut using Cutadapt (v1.9.1, http://cutadapt.readthedocs. io/en/stable/), and then each sample data was separated from the obtained reads according to the Barcode. The Barcode and primer sequence were cut off to obtain Raw Reads. The Reads sequence was aligned with the species annotation database through https://github.com/torognes/vsearch/ (*Mahé and Rognes, 2025*) to detect chimera sequences, and finally the chimera sequences were removed to obtain the Clean Reads. The sequences were clustered into Operational Taxonomic Units (OTUs) with 97% identity using Uparse software (v7.0,1001, http://www.drive5.com/uparse/) for all Clean Reads of the sample. Species annotation of OTU sequences was performed using the Mothur method with the SSUrRNA database of Silva 132 (http://www.arb-silva.de/), and multiple sequence alignment was performed using MUSCLE (v3.8.31, http://www.drive5.com/muscle/) software to obtain the phylogenetic relationships of all OTUs sequences. The data of each sample were normalized using a standard of sequence number corresponding to the sample with the least sequences. Subsequent analysis of alpha diversity and beta diversity were all performed based on the output normalized data. Alpha diversity is applied in analyzing complexity of species diversity for a sample through four indices, including Chao1, Observed species, Shannon, Simpson, and all these indices in our samples were calculated with QIIME (v1.9.1) and analyzed by R software (v2.15.3) using T-test and Benjamini-Hochberg. Unifrac distance and the UPGMA sample clustering tree were calculated by QIIME (v1.9.1), and the Anosim, principal coordinate analysis (PCoA), and non-metric multidimensional scaling (NMDS) analyses were drawn and implemented through R software (v2.15.3) with

anosim, WGCNA, stats, ggplot2 and vegan packages. Linear discriminant analysis (LDA) effect size (LEfSe) package was used for LEfSe analysis. Other diagrams were implemented by using the R software (v2.15.3).

## Quantification of short chain fatty acids

The colonic contents (50 mg) were homogenized with 0.5 ml dH$_2$O in ball mill and then centrifuged. The supernatant 1 (0.3 ml) was homogenized with 0.5 ml dH$_2$O and then centrifuged. The supernatant 2 (0.5 ml) combined with supernatant 1 were mixed with 0.8 ml of 2-Methylvaleric acid as internal standard solution and 0.1 ml 50% H$_2$SO$_4$. After centrifuging, the supernatant was transferred into fresh glass vial for Gas chromatography-mass spectrometry analysis, which was performed using an Agilent 7890B gas chromatograph system coupled with an Agilent 5977B mass spectrometer. Short-chain fatty acids (SCFA) were identified and quantified by the retention time and standard curves of the standard solutions.

## Flow cytometry (FC) analysis

Intestinal lamina propria cells were isolated by using the Mouse Lamina Propria Dissociation Kit (Miltenyi Biotec, Germany) according to the instruction. For cell surface staining, the cells were incubated with FC block at 2–8°C for 15 min, then were stained with Live/Dead Dye (FVS510, BD Pharmingen, USA) at room temperature for 15 min, followed by washing with Stain Buffer. Cells were incubated with specific fluorescent antibodies (*Supplementary file 1d*) at 2–8°C for 30 min to label immune cells. For intracellular factor staining, cells were fixed and permeabilized using Transcription Factor Fix/Prem Buffer at 4°C for 40 min, washed with 1×Perm/Wash Buffer, and stained with intracellular markers at 4°C for 30 min. The stained cells were washed and resuspended in PBS, and assessed by the 12-color FACSCelesta flow cytometer (Becton, Dickinson and Company, USA). The single cells were first gated on SSC-A vs FSC-A and FSC-A vs FSC-H, and then CD45$^+$Live$^+$ cells were gated on single cells. The CD45$^+$CD3$^-$ cells were gated on CD45$^+$Live$^+$ cells. The CD4$^+$ T cells (CD3$^+$CD4$^+$ cells) were gated on CD45$^+$CD3$^-$ cells, Tregs (CD25$^+$Foxp3$^+$ cells), and Th17 cells (CD4$^+$RORt$^+$ cells) were gated on CD3$^+$CD4$^+$ cells. The CD86$^+$CD11C$^+$ cells were gated on DC$^+$CD3$^-$ cells. The data were analyzed using FlowJo software (v10.8.0).

## Isolation and culture of mouse intestinal crypts

The jejunum proximal to the stomach was harvested immediately after mice were sacrificed. After gently flushing with cold D-PBS (without calcium and magnesium) using injection syringe, the intestinal segment was opened longitudinally and gently washed with cold D-PBS three times. Then, the washed intestine was cut into 2 mm pieces with scissors, which fell into a 50 ml conical tube containing 15 ml cold D-PBS. The rinsing procedure using the 10 ml serological pipette to gently pipette the intestinal pieces up and down three times and then aspirate off the supernatant and add 15 ml fresh cold D-PBS was repeated 10-15 times until the supernatant was clear. The intestinal pieces were resuspended in 25 ml Gentle Cell Dissociation Reagent (GCDR, StemCell Technologies, Canada) and incubated at room temperature for 15 min on a rocking platform. After incubation, the intestinal pieces were resuspended in 10 ml cold D-PBS containing 0.1% BSA, supernatant was filtered through a 70 μm filter (Corning, USA), and filtrate was collected and labeled as fraction 1. Repeat the above step three times to obtain fractions 2-4. The fractions were centrifuged at 290×g for 5 min at 4°C and resuspended in 10 ml cold DMEM/F-12 with 15 mM HEPES. The quality of the fractions was assessed by using an inverted microscope. The fraction which enriched for intestinal crypts was selected, centrifuged, and resuspended in equal volumes of complete IntestiCult Organoid Growth Medium (06005, StemCell Technologies, Canada) and Matrigel Matrix. The 50 μl of the 500-crypt suspension was plated in the center of each well of the pre-warmed 24-well plate. The plate was placed at 37°C with 5% CO$_2$ for 10 min and then added 750 μl complete IntestiCult Organoid Growth Medium containing 100 μg ml$^{-1}$ penicillin/streptomycin to each well. The culture medium was fully exchanged three times per week. After 7 to 10 d of culture, the organoids were passaged using 1:6 split ratio. The images of organoids were captured using inverted microscope (Nikon, Japan). The surface areas of organoids were measured using ImageJ software (v1.8.0). The forming efficiency (%) = (number of mature organoids growing after 5 d/number of crypts seeded)×100%.

## Statistical analysis

Data were statistically analyzed by SPSS (v26.$^0$) software, and all data are presented as means ± standard deviation (SD). p-value < 0.05 was considered significant. Kolmogorov-Smirnov test was used to determine whether the data followed the normal distribution. Comparisons between two groups were executed by unpaired Student's t-test or Mann-Whitney U-test, and comparisons among three groups were executed by one-way ANOVA or Kruskal-Wallis followed by Dunn's multiple comparisons.

## Acknowledgements

We appreciate the Experimental Teaching Center, College of Animal Science, Zhejiang University for providing the AW400SG anaerobic workstations. This research was supported by the National Natural Science Foundation of China (grant number 32172765).

## Additional information

### Funding

| Funder | Grant reference number | Author |
| --- | --- | --- |
| National Natural Science Foundation of China | 32172765 | Xinyan Han |

The funders had no role in study design, data collection and interpretation, or the decision to submit the work for publication.

### Author contributions

Xin Ma, Writing – original draft, Writing – review and editing, Carried out all aspects of experiments and collected the data; Meng Li, Validation; Yuanyuan Zhang, Methodology; Tingting Xu, Supervision; Xinchen Zhou, Assisted with preparing the manuscript; Mengqi Qian, Assisted with immunofluorescence experiments and data analysis; Zhiren Yang, Software; Xinyan Han, Funding acquisition, Project administration

### Author ORCIDs

Xinyan Han  https://orcid.org/0000-0001-7044-7439

Reviewer #3 (Public review): https://doi.org/10.7554/eLife.92906.5.sa1
Author response https://doi.org/10.7554/eLife.92906.5.sa2

## Additional files

### Supplementary files

Supplementary file 1. Supplementary tables (1 a–1d) described throughout main text.

### Data availability

The contigs for all samples derived from microbiome analysis can be found in the Sequence Read Archive (SRA) (https://www.ncbi.nlm.nih.gov/sra/PRJNA837047). And all data generated or analysed during this study are included in the manuscript and supporting files; source data files have been provided for Figures 3, 5, 7, 9 and 10.

The following dataset was generated:

| Author(s) | Year | Dataset title | Dataset URL | Database and Identifier |
| --- | --- | --- | --- | --- |
| Ma X | 2022 | The sequencing of gut microbiota | https://www.ncbi.nlm.nih.gov/sra/PRJNA837047 | NCBI Sequence Read Archive, PRJNA837047 |

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
