## [Editor Report · eLife Assessment]

The work by Han and collaborators describes **valuable** findings on the role of Akkermansia muciniphila during ETEC infection. If confirmed, these findings will add to a growing list of beneficial properties of this organism. The strength of the evidence used to justify the conclusions in the manuscript is **solid**, as the analyses broadly support the claims with only minor weaknesses.

---

## [Referee Report · Reviewer #3 (Public review)]

Summary:

The manuscript by Ma et al. describes a multi-model (pig, mouse, organoid) investigation into how fecal transplants protect against *E. coli* infection. The authors identify A. muciniphila and B. fragilis as two important strains and characterize how these organisms impact the epithelium by modulating host signaling pathways, namely the Wnt pathway in lgr5 intestinal stem cells.

Strengths:

The strengths of this manuscript include the use of multiple model systems and follow up mechanistic investigations to understand how A. muciniphila and B. fragilis interacted with the host to impact epithelial physiology.

Weaknesses:

As in previous revisions, there remains concerning ambiguity in the methodology used for microbiota sequence analysis and it would be difficult to replicate the analysis in any meaningful way. In this revision, concerns about the rigor and reproducibility of this component of the manuscript have been increased. Readers should be cautious with interpretation of this data.

(1) In previous versions of the manuscript it would appear the correct bioproject accession was listed but, the actual link went to an unrelated project. The updated accession link appears to contain raw data; however, the authors state they used an Illumina HiSeq 2500. This would be an unusual choice for V3-V4 as it would not have read lengths long enough to overlap. Inspection of the first sample (SRR19164796) demonstrates that this is absolutely not the raw data, as there is a ~400 nt forward read, and a 0 length reverse read. All quality scores are set to 30. There is no logical way to go from HiSeq 2500 raw data and read lengths to what was uploaded to the SRA and it was certainly not described in the manuscript.

(2) No multiple testing correction was applied to the microbiome data.

---

## [Author Response]

The following is the authors’ response to the current reviews.

Those comments are all valuable and very helpful for revising and improving our paper, as well as the important guiding significance to our researches. We have studied comments carefully and have made correction which we hope meet with approval.

**Reviewer #3 (Public review):**
Summary:The manuscript by Ma et al. describes a multi-model (pig, mouse, organoid) investigation into how fecal transplants protect against *E. coli* infection. The authors identify *A. muciniphila* and B. fragilis as two important strains and characterize how these organisms impact the epithelium by modulating host signaling pathways, namely the Wnt pathway in lgr5 intestinal stem cells.Strengths:The strengths of this manuscript include the use of multiple model systems and follow up mechanistic investigations to understand how *A. muciniphila* and *B. fragilis* interacted with the host to impact epithelial physiology.Weaknesses:As in previous revisions, there remains concerning ambiguity in the methodology used for microbiota sequence analysis and it would be difficult to replicate the analysis in any meaningful way. In this revision, concerns about the rigor and reproducibility of this component of the manuscript have been increased. Readers should be cautious with interpretation of this data.(1) In previous versions of the manuscript it would appear the correct bioproject accession was listed but, the actual link went to an unrelated project. The updated accession link appears to contain raw data; however, the authors state they used an Illumina HiSeq 2500. This would be an unusual choice for V3-V4 as it would not have read lengths long enough to overlap. Inspection of the first sample (SRR19164796) demonstrates that this is absolutely not the raw data, as there is a ~400 nt forward read, and a 0 length reverse read. All quality scores are set to 30. There is no logical way to go from HiSeq 2500 raw data and read lengths to what was uploaded to the SRA and it was certainly not described in the manuscript.

What we uploaded to the SRA was Contigs files for sample, we have modified the description on line 694.

(2) No multiple testing correction was applied to the microbiome data.

The alpha diversity indexes were tested using T-test and wilcox test, and we showed the result of T-test in FigureS1B. The p-values were corrected for multiple testing using the Benjamini-Hochberg method, we have modified the description on line 322.

---------

The following is the authors’ response to the previous reviews.

**Public Reviews:**

**Reviewer #2 (Public Review):**
Ma X. et al proposed that *A. muciniphila* was a key strain that promotes the proliferation and differentiation of intestinal stem cells through acting on the Wnt/β-catenin signaling pathway. They used various models, such as piglet model, mouse model and intestinal organoids to address how *A. muciniphila* and *B. fragilis* offer the protection against ETEC infection. They showed that FMT with fecal samples, *A. muciniphila* or *B. fragilis* protected piglets and/or mice from ETEC infection, and this protection is manifested as reduced intestinal inflammation/bacterial colonization, increased tight junction/Muc2 proteins, as well as proper Treg/Th17 cells. Additionally, they demonstrated that *A. muciniphila* protected basal-out and/or apical-out intestinal organoids against ETEC infection via Wnt signaling.Comments on revised version:Please add proper references to indicate the invasion of ETEC into organoids after 1 h of infection.

We have added references on line 211.

References:

Xiao K, Yang Y, Zhang Y, Lv QQ, Huang FF, Wang D, Zhao JC, Liu YL. 2022. Long-chain PUFA ameliorate enterotoxigenic *Escherichia coli*-induced intestinal inflammation and cell injury by modulating pyroptosis and necroptosis signaling pathways in porcine intestinal epithelial cells. *Br. J. Nutr.* 128(5):835-850.

Qian MQ, Zhou XC, Xu TT, Li M, Yang ZR, Han XY. 2023. Evaluation of Potential Probiotic Properties of *Limosilactobacillus fermentum* Derived from Piglet Feces and Influence on the Healthy and *E. coli*-Challenged Porcine Intestine. *Microorganisms*. 11(4).

**Reviewer #3 (Public Review):**
Summary:The manuscript by Ma et al. describes a multi-model (pig, mouse, organoid) investigation into how fecal transplants protect against *E. coli* infection. The authors identify *A. muciniphila* and *B. fragilis* as two important strains and characterize how these organisms impact the epithelium by modulating host signaling pathways, namely the Wnt pathway in lgr5 intestinal stem cells.Strengths:The strengths of this manuscript include the use of multiple model systems and follow up mechanistic investigations to understand how *A. muciniphila* and *B. fragilis* interacted with the host to impact epithelial physiology.Weaknesses:After an additional revision, the bioinformatics section of the methods has changed significantly from previous versions and now indicates a third sequencer was used instead: Ion S5 XL. Important parameters required to replicate analysis have still not been provided. Inspection of the SRA data indicates a mix of Illumina MiSeq and Illumina HiSeq 2500. It is now unclear which sequencing technology was used as authors have variably reported 4 different sequencers for these samples. Appropriate metadata was not provided in the SRA, although some groups may be inferred from sample names. These changing descriptions of the methodologies and ambiguity in making the data available create concerns about rigor of study and results.

Due to confusing the sequencing method of this experiment with other experiment samples, we apologize for the multiple incorrect modifications of the method description. We have modified the method for microbiome sequencing technology on line 304. The sequencing technology is Illumina HiSeq 2500. The SRA metadata can be viewed at https://www.ncbi.nlm.nih.gov/sra/PRJNA837047. The sample names ep1-6 and ef1-6 were correspond to the EP and EF groups, respectively.

**Recommendations For the Authors:**
As in the previous revision:-provide important parameters required to replicate analysis-ensure that reporting of sequencing technology is correct as data listed on SRA appears to be derived from Illumina sequencers, and was deposited indicating as such.-update SRA metadata such that experimental groups are clear and match the nomenclature used in the manuscript Particularly for samples which are labelled [A-Z][0-9]

- The multiple testing correction wasn’t applied.

-Due to confusing the sequencing method of this experiment with other experiment samples, we apologize for the multiple incorrect modifications of the method description. We have modified the method for microbiome sequencing technology on line 304. The sequencing technology is Illumina HiSeq 2500.

- The SRA metadata can be viewed at https://www.ncbi.nlm.nih.gov/sra/PRJNA837047. The sample names ep1-6 and ef1-6 were correspond to the EP and EF groups, respectively.